# Neuronal and glial DNA methylation and gene expression changes in early epileptogenesis

**Toni C. Berger**[ID][1,2ʘ]\*, **Magnus D. Vigeland**[3ʘ], **Hanne S. Hjorthaug**[3], **Lars Etholm**[4,5], **Cecilie G. Nome**[2], **Erik Taubøll**[1,2], **Kjell Heuser**[1,2‡], **Kaja K. Selmer**[3,4,6‡]

**1** Department of Neurology, Oslo University Hospital, Oslo, Norway, **2** University of Oslo, Oslo, Norway, **3** Department of Medical Genetics, Oslo University Hospital and University of Oslo, Oslo, Norway, **4** National Center for Epilepsy, Oslo University Hospital, Sandvika, Norway, **5** Department of Neurology, Section for Neurophysiology, Oslo University Hospital, Oslo, Norway, **6** Division of Clinical Neuroscience, Department of Research and Development, Oslo University Hospital, Oslo, Norway

ʘ These authors contributed equally to this work.
‡ KH and KKS also contributed equally to this work.
\* toni.berger@medisin.uio.no

## Abstract

### Background and aims

Mesial Temporal Lobe Epilepsy is characterized by progressive changes of both neurons and glia, also referred to as epileptogenesis. No curative treatment options, apart from surgery, are available. DNA methylation (DNAm) is a potential upstream mechanism in epileptogenesis and may serve as a novel therapeutic target. To our knowledge, this is the first study to investigate epilepsy-related DNAm, gene expression (GE) and their relationship, in neurons and glia.

### Methods

We used the intracortical kainic acid injection model to elicit status epilepticus. At 24 hours post injection, hippocampi from eight kainic acid- (KA) and eight saline-injected (SH) mice were extracted and shock frozen. Separation into neurons and glial nuclei was performed by flow cytometry. Changes in DNAm and gene expression were measured with reduced representation bisulfite sequencing (RRBS) and mRNA-sequencing (mRNAseq). Statistical analyses were performed in R with the edgeR package.

### Results

We observed fulminant DNAm- and GE changes in both neurons and glia at 24 hours after initiation of status epilepticus. The vast majority of these changes were specific for either neurons or glia. At several epilepsy-related genes, like *HDAC11*, *SPP1*, *GAL*, *DRD1* and *SV2C*, significant differential methylation and differential gene expression coincided.

### Conclusion

We found neuron- and glia-specific changes in DNAm and gene expression in early epileptogenesis. We detected single genetic loci in several epilepsy-related genes, where DNAm

**Data Availability Statement:** Raw data are available from: https://www.ncbi.nlm.nih.gov/geo/query/acc.cgi?acc=GSE138100.

**Funding:** This project has participated in the European Commission, ERA-NET NEURON, Brain Inflammation, Glia and Epilepsy (K.H.), and has received funding from the European Union's Horizon 2020 research and innovation programme under the Marie Sklodowska-Curie grant agreement No 722053 (to K.H.). The project was also funded from South-Eastern Norway Regional Health Authority, No 2014018 (K.K.S.). The funders had no role in study design, data collection and analysis, decision to publish, or preparation of the manuscript.

**Competing interests:** The authors have declared that no competing interests exist.

and GE changes coincide, worth further investigation. Further, our results may serve as an information source for neuronal and glial alterations in both DNAm and GE in early epileptogenesis.

## Introduction

Epilepsy is defined as an inherent predisposition of the brain to recurrently generate epileptic seizures [1] and affects an estimated 65 million people world-wide [2]. Temporal lobe epilepsy (TLE) is the most common subtype amongst the focal epilepsies [3], with hippocampal sclerosis being detected in 70% of drug resistant TLE patients [4, 5]. The typical clinical course of the sub-entity, mesial temporal lobe epilepsy with hippocampal sclerosis (mTLE-HS), is characterized by an initial precipitating event (e.g. cerebral trauma, inflammation, prolonged febrile seizure), a seizure-free latency period and finally the onset of spontaneous and progressive seizures [6]. This metamorphosis into a brain prone to spontaneous recurrent seizures of progressive nature, also referred to as epileptogenesis [7, 8], is characterized by a plethora of cellular and molecular changes in both neurons and glia [5, 9–18].

One third of people with epilepsy respond inadequately to treatment with the primarily symptom-alleviating antiepileptic drugs of today [19], rendering the identification of potential upstream effectors of epileptogenesis and the development of disease modifying antiepileptic drugs a task of upmost importance [20, 21].

DNAm, in the context of this paper, the methylation of CpG nucleotides in the DNA [22], plays a primordial role in brain development, cell fate, tissue specific gene expression [22–25]. It has further been shown to be modified by neuronal activity [26]. Alterations of DNAm in epileptogenesis encompass upregulation of DNA–methyl–transferases, enzymes methylating the DNA base cytosine, in human TLE patients [27], genome wide changes in DNAm during epileptogenesis [28, 29], and a later onset of spontaneous seizures in murine epilepsy models under treatment with a DNA-methyl-transferase inhibitor [30].

With the dawn of new site- and cell specific epigenetic modulators like modified CRISPR, zinc finger proteins and transcription-activator-like-effectors [31–33], the identification of potential genomic sites for antiepileptogenic intervention is of utmost importance.

DNAm in neurons and glial cells is mostly cell specific [34, 35]. Sorting of brain tissue into specific cell types prior to downstream analysis has been applied in previous studies of gene expression [36–38] and DNA methylation [39–41]. This approach provides information about the cellular origin of observed effects on the epigenome and transcriptome level and an elevated detection sensitivity of more subtle changes in DNAm and GE.

Hypotheses for this study are i) that epileptogenesis affects DNAm and GE in a cell specific manner and ii) that differential methylation (DM) in neurons and glial cells correlates with differential gene expression (DGE). To our knowledge, this is the first study to investigate DNAm and GE changes as well as their possible association in neurons and glia separately, and the first one of its kind conducted in the widely used intracortical mouse model of mTLE [15].

## Methods

### Animals

Adult male C57/BL6N mice (Janvier lab) were acquired at an age of 8 weeks, acclimatized for 4 weeks in a controlled environment (21-23˚C, 12h dark/light cycles), 1–4 animals per cage,

with water and food available *ad libitum*. All animal procedures were approved by the Norwegian Food Safety Authority (national ethics committee, project number FOTS: 14198) and the Centre for Comparative Medicine, Oslo University Hospital and the University of Oslo.

## Intracortical kainic acid mouse model of mTLE

We used deep cortical (juxta hippocampal) kainic acid injection to elicit an initial status epilepticus. The animal model has been described in detail in a separate paper [15]. Briefly, mice injected with kainic acid typically (91%) develop chronic epilepsy in a stagewise manner. During the acute stage directly after kainic acid injection, animals undergo a status epilepticus lasting around 4 hours (4.4 +- 2.4 hrs). This stage is followed by a clinically silent latent phase that lasts around 5 days (5+-2.9 days). The first spontaneous seizure at the end of this stage also marks the start of the last, chronic, stage of epileptogenesis, characterized by spontaneous seizures of progressive nature. For kainic acid injections, mice were anesthetized with a mixture of medetomidine (0.3 mg/kg, i.p.) and ketamine (40 mg/kg, i.p.) and kept on a heating blanket. A small craniotomy was performed in a stereotactic frame above the right hippocampus. Then kainic acid (70 nl, 20 mM, Tocris) was injected by a Hamilton pipette (Hamilton Company, NV) at a depth of 1.7 mm at the following coordinates relative to Bregma: anteroposterior −2 mm, lateral +1.5 mm (right). Anesthesia was stopped with atipamezole (300 mg/kg, i.p.). All mice received buprenorphine (0.1mg/kg, s.c.) at 4 and 12 hours after the intervention. Animals in the KA group not displaying convulsive seizures were excluded from further analysis. SH animals underwent the same procedures as described for the KA group, apart from 0,9% NaCl used instead of kainic acid for the intracortical injection.

## Tissue collection and pooling

At 24 hours after status epilepticus, cervical dislocation was performed under anesthesia and right hippocampi were extracted. After extraction, each hemisphere was placed in a 2 mL polypropylene tube, instantly shock frozen in liquid nitrogen, and stored at -80˚C. Right hippocampi were pooled in 2 mL tubes from 4 (KA n = 4, S n = 4) or 2 (KA n = 4, S n = 4) mice prior to further processing. The number of mice per group (KA, SH) amounted to 8 mice per group, the number of biological samples to 3 per group (KA, SH). Tissue was kept on dry ice during pooling. See also S1 Fig.

## Fluorescent Activated Nuclear Sorting (FANS)

A modified version of a nuclear sorting protocol by Jiang *et al*. [42] was used to sort into NeuN+ (refered to as neurons) and NeuN- (refered to as glia) nuclei. Immediately after pooling, hippocampi were placed on ice, and 1 mL homogenization buffer was added to each pool. Tissue was homogenized using a GentleMACS dissociator (Miltenyi), and homogenate filtered through a 70 μm filter. A NeuN-negative control sample from adult mouse liver was processed in parallel with the hippocampal samples. Debris was removed by density gradient centrifugation, using Debris Removal Solution (Miltenyi), and nuclear pellets were resuspended in 100 μL incubation buffer per one million nuclei. Anti-NeuN Alexa Fluor488 (Merck Millipore) was added to each sample to a final concentration of 0.1 μg/mL, and samples incubated for 1 h on ice in a light protected environment. Sorting of nuclei was performed on a FACSAria (BD Biosciences). Propidium iodide (PI) was added prior to sorting, and the following strategy was used for gating (S2 Fig): 1) A nuclear gate was defined by PI-positive events. 2) Aggregated nuclei were excluded in a dot plot using the pulse width of side scatter (SSC-w) versus the pulse area of forward scatter (FSC-a). 3) NeuN-negative gate was drawn based on signal from anti-NeuN stained liver sample. NeuN-positive and NeuN-negative hippocampi nuclei were

sorted into tubes, and nuclei pelleted by centrifugation. Pellets were resuspended in lysis buffer for downstream DNA and RNA isolation. A full description of the FANS procedure is given in S1 Supporting Information.

### Isolation of DNA and total RNA from sorted nuclei

DNA was extracted from sorted nuclei with MasterPure Complete DNA and RNA Purification Kit (Epicentre), DNA purity was assessed on NanoDrop, and DNA concentration measured on Qubit (DNA HS assay). Further details are available in S1 Supporting Information. Lysates were thawed on ice and total RNA extracted with mirVana miRNA Isolation Kit (Ambion). Up-concentration was performed using RNA Clean & Concentrator-5 kit (Zymo Research). RNA concentration and integrity were assessed on Bioanalyzer with the RNA Pico Kit (Agilent Technologies). Further details are found in S1 Supporting Information.

### RRBS

A modified version of the gel-free protocol by Boyle *et al.* [43] was used for RRBS library prep-aration. The main changes to the protocol were inclusion of a two-sided size selection prior to bisulfite conversion, and sample pooling performed after completion of single libraries. A full description of the RRBS library prep and sequencing is given in S1 Supporting Information. Libraries were subjected to either 75 bp single read sequencing on NextSeq500 (Illumina), or 150 bp single read sequencing on HiSeq2500 (Illumina). For sequencing on NextSeq500, a pool of 14 libraries were run twice, with 50% PhiX spike-in at each run. On HiSeq2500, pools of 15 libraries were sequenced over two lanes, using 10% PhiX spike-in.

### High throughput mRNAseq

SMART-Seqv4 Ultra Low InputRNA Kit for Sequencing (Takara Bio) was used to amplify mRNA from total RNA, and the resulting cDNA was used as input in library preparation with ThruPlex DNAseq Kit (Rubicon Genomics). See S1 Supporting Information for details regard-ing cDNA synthesis and library preparation. Libraries were sequenced on NextSeq500 (75 bp single read), or HiSeq3000 (150 bp paired end). On NextSeq500, 12 libraries were pooled for one sequencing-run. The remaining 27 libraries were sequenced in one pool over three lanes on HiSeq3000.

### Computational methods

**RRBS- and mRNAseq-post processing.** Post-processing included trimming of reads using Trim Galore! v0.4.3 with parameters "—rrbs—illumina" and quality control with FastQC. Alignment to the reference mouse genome mm10 was performed with Bismark v0.20, powered by bowtie2. Quality metrics were collected from the resulting BAM files using the Picard tool CollectRrbsMetrics v2.18.15.

Alignment of the mRNAseq reads was accomplished with the Subread package through its R interface Rsubread, after trimming with Trim Galore! v0.4.3. Quality control of the resulting BAM files was undertaken with CollectRnaSeqMetrics from Picard v2.18.15. Uniquely mapped reads were assigned to genes and counted by the featureCounts function of Rsubread, using default parameters. As reference for the gene assignment we used release M16 of the compre-hensive gene annotation of mm10 available from GENCODE. Only RNA aligning to mRNA regions was used for further analysis.

**Annotation.** The mouse genome build mm10 was used as reference in all analyses. Only autosomal data were analyzed. Coordinates of genes, exons and introns were taken from

GENCODE's comprehensive annotation (www.gencodegenes.org/mouse/release_M16.html). The R package annotatr [44] was used to bioinformatically link CpG sites to the gene annotations.

Using predefined genomic features within annotatr [44], the promoter region for any gene was defined as the segment from -1kb upstream to the transcription start site and the upstream region from -5 kb to -1 kb, where negative numbers indicate positions upstream of transcription start site.

## Statistical methods

**Analysis of DGE.**   The R package edgeR [45] was used to identify differentially expressed genes in the mRNAseq (mRNA) data set. The data from neuronal and glia cells were treated separately, contrasting KA versus SH samples in each case. Genes without official HGNC symbol were excluded from the analysis. Genes with low read counts were also removed, using the edgeR function filterByExpr with default parameters. After normalization to adjust for different library sizes (calcNormFactors) we followed a standard edgeR workflow to fit a quasi-likelihood negative binomial generalized log-linear model to the count data, and to perform the subsequent statistical analysis. A false discovery rate (FDR) approach was adopted to account for multiple testing, with a significance threshold of FDR 25%.

**Analysis of DM.**   To identify loci exhibiting differential methylation between KA and SH samples, we adopted the edgeR workflow for RRBS data recently published by the edgeR authors [46]. Briefly, this approach entails treating the methylated and unmethylated counts at each locus as independent variables following a negative binomial distribution. As in the DGE analysis, DM analysis was carried out separately for neurons and glia cells, with an FDR of 25% as threshold for statistical significance. Before the analysis, filters were applied to all CpG sites where more than 10% of the samples had either very low coverage ($<$ 8 reads) or excessively high coverage ($>$ 99.5 quantile across all sites and samples). The DM analysis was performed both at the level of individual CpG sites, and in aggregated form within pre-defined genomic features: upstream, promoter, UTR5, exons, introns, gene body (i.e. the union of all exons and introns of a specific gene) and UTR3. For the aggregated analysis the input was the mean counts across all covered CpG's within the region.

**Combined DM and DGE analysis.**   For each genomic feature (upstream, promoter, UTR5, exon, intron, gene body, UTR3), a combined analysis of DGE and DM was performed in order to unveil genes for which both methylation and gene expression differed significantly between the two groups. To reduce the statistical noise, the DGE analysis was reanalyzed for each genomic feature type, using only the relevant subset of the data. Specifically, for each genomic feature type, only the genes present in the aggregated DM data set were kept in the DGE analysis. Co-incidence of DGE and DM was declared for features surviving an FDR cut-off of 25% in both analyses.

**Functional enrichment analysis.**   Enrichment analyses of Gene Ontology (GO) and Kyoto Encyclopedia of Genes and Genomes (KEGG) pathways were performed with the goana and kegga functions of edgeR, with the parameter species = "Mm". These functions conduct overlap tests for the up- and down-regulated DE genes, and for the genes overlapping DMRs.

## Quality control

**Bisulfite conversion rate estimation.**   The conversion rate estimate computed by Picard/ CollectRrbsMetrics is based on the conversion of non-CpG cytosines. As methylation of non-CpG cytosines is non-negligible in neuronal cells, this may bias the results. To account for this,

we also performed an alternative estimate of the conversion rates directly from the untrimmed fastq files, by checking the methylation status specifically on the (unmethylated) cytosines added in the end-repair step of the RRBS preparation (private bash script). See S1 Supporting Information for further information of bisulfite conversion rate.

**Multidimensional scaling.** In order to validate our cell sorting procedures, and look for outliers among the samples, multidimensional scaling (MDS) plots were produced for the mRNAseq and RRBS data sets. The MDS computations were done by the plotMDS function of edgeR, selecting the top 100 most variable loci. The actual plots were created with ggplot2 [47].

**Expression of neuronal and glial genes in NeuN+ and NeuN- fraction.** Normalized counts for expression of neuronal (*RBFOX3*), glial (*ALDHL1L1*, *CX3CR1*, *MBP*), *pericytal (PDGFRB) and endothelial (PECAM1)* genes were used to visualize enrichment of neurons in the NeuN+ fraction and glia in the NeuN- fraction.

## Selection of relevant GO and KEGG terms

Relevant (Tables 1 and 2) and epilepsy-relevant (Fig 2) GO and KEGG terms in neurons and glia were selected manually based on reviews on the subject [9] and personal knowledge. The list of GO and KEGG terms derived from our DGE analysis (for a full list see S1 Table) was manually filtered for specific terms and relevant up-/downregulated genes within these terms in neurons and glia selected for presentation.

# Results

## Quality control

Bisulfite conversion rates were above 98% (S1 Supplementary Information and S3 Fig) and multidimensional scaling plots of mRNAseq and RRBS data sets distinguished clearly between neurons and glia (S4 Fig and S5 Fig). The NeuN+ fraction enriched for neuronal and the NeuN- fraction for glial mRNA (S6 Fig), indicating a successful separation of neurons and glia.

## Differential methylation

A statistical analysis of Differentially Methylated CpGs (DM CpGs) compared right hippocampi of KA to SH mice at 24 hours post injection. After filtering, 928 430 CpG sites remained and were used in subsequent analyses. On average, across all CpG sites and all samples, each CpG was covered by 29.8 reads. In individual samples the mean read depth varied from 20.0 to 35.2 (median = 30.6, interquartile range = [27.4–32.7]).

**Differentially methylated sites.** The analysis of significantly altered DM CpGs revealed 1060 hyper- and 899 hypomethylated (ratio 1.2:1) CpG sites in neurons and 464 hyper- and 274- hypomethylated (ratio: 1.7:1) CpG sites in glia (Fig 1). Most of the DM CpGs localized to either gene bodies or intergenic regions (for full list see supplementary 2) and were distributed evenly across chromosomes (sex chromosomes excluded), apart from a possible higher ratio of hyper-/hypomethylated CpGs at chromosome 13 to 15 in glia. The ratio of hypermethylated to hypomethylated CpG sites was highest at upstream (1.3:1) and intergenic (1.2:1) regions for neurons and upstream (2.7:1) and promoter (2.0:1) for glia. For detailed information including GO and KEGG annotation of DM CpGs see S1 Table.

**Differentially methylated CpG sites common to both neurons and glia.** Neurons and glia shared four commonly hypermethylated (fraction: 0.3%) and zero commonly hypomethylated DM CpGs. One DM CpG was hypermethylated in neurons and hypomethylated in glia (fraction: 0.0%) and one DM CpGs hypomethylated in neurons and hypermethylated in glia (fraction: 0.0%). Three of the four commonly hypermethylated DM CpGs localized to gene

**Table 1. Differentially expressed genes in neurons at 24 hours post injection in the intracortical kainic acid model of mTLE.**

| Differentially expressed genes in neurons | | | |
|---|---|---|---|
| Upregulated genes (N = 135) | | | |
| Gene symbol | logFC | FDR | Gene description |
| Acan | 3,94 | 0,000 | aggrecan |
| Sdc1 | 3,60 | 0,001 | syndecan 1 |
| Inhba | 4,12 | 0,001 | inhibin beta-A |
| Socs3 | 4,21 | 0,001 | suppressor of cytokine signaling 3 |
| Timp1 | 5,38 | 0,007 | tissue inhibitor of metalloproteinase 1 |
| Megf11 | 2,25 | 0,011 | multiple EGF-like-domains 11 |
| Nptx2 | 3,61 | 0,011 | neuronal pentraxin 2 |
| Hspa1a | 5,46 | 0,011 | heat shock protein 1A |
| Fgl2 | 3,07 | 0,011 | fibrinogen-like protein 2 |
| Col27a1 | 3,22 | 0,012 | collagen, type XXVII, alpha 1 |
| Mapk4 | 2,28 | 0,012 | mitogen-activated protein kinase 4 |
| Cd1d1 | 2,94 | 0,018 | CD1d1 antigen |
| Sik1 | 2,31 | 0,019 | salt inducible kinase 1 |
| Hspa1b | 4,82 | 0,019 | heat shock protein 1B |
| Tnc | 2,08 | 0,019 | tenascin C |
| Ptgs2 | 3,11 | 0,019 | prostaglandin-endoperoxide synthase 2 |
| Gipr | 3,72 | 0,019 | gastric inhibitory polypeptide receptor |
| Trib1 | 2,46 | 0,019 | tribbles pseudokinase 1 |
| Tpbg | 2,10 | 0,019 | trophoblast glycoprotein |
| Lhfp | 1,71 | 0,019 | lipoma HMGIC fusion partner |
| Fosb | 3,32 | 0,019 | FBJ osteosarcoma oncogene B |
| Arc | 2,40 | 0,020 | activity regulated cytoskeletal-associated protein |
| Fosl2 | 2,26 | 0,020 | fos-like antigen 2 |
| Gadd45g | 2,54 | 0,027 | growth arrest and DNA-damage-inducible 45 gamma |
| Pcdh11x | 2,26 | 0,027 | protocadherin 11 X-linked |
| Pmepa1 | 2,30 | 0,027 | prostate transmembrane protein, androgen induced 1 |
| Stk40 | 2,03 | 0,027 | serine/threonine kinase 40 |
| Pde6b | 2,91 | 0,027 | phosphodiesterase 6B, cGMP, rod receptor, beta polypeptide |
| Wisp1 | 2,09 | 0,027 | WNT1-inducible-signaling pathway protein 1 |
| 9330188P03Rik | 3,35 | 0,027 | RIKEN cDNA 9330188P03 gene |
| Pappa | 2,89 | 0,027 | pregnancy-associated plasma protein A |
| Hspb1 | 4,00 | 0,027 | heat shock protein 1 |
| Atf3 | 4,52 | 0,029 | activating transcription factor 3 |
| Tll1 | 3,71 | 0,029 | tolloid-like |
| Sulf1 | 1,57 | 0,031 | sulfatase 1 |
| Lbh | 3,01 | 0,034 | limb-bud and heart |
| Nedd9 | 1,51 | 0,035 | neural precursor cell expressed, developmentally down-regulated gene 9 |
| Parp3 | 2,93 | 0,035 | poly (ADP-ribose) polymerase family, member 3 |
| Rrad | 4,89 | 0,035 | Ras-related associated with diabetes |
| Trh | 6,22 | 0,035 | thyrotropin releasing hormone |
| 4931440P22Rik | 1,70 | 0,037 | RIKEN cDNA 4931440P22 gene |
| Cyr61 | 2,90 | 0,037 | Cysteine-rich angiogenic inducer 61 |
| Fos | 2,89 | 0,037 | FBJ osteosarcoma oncogene |
| Cgref1 | 2,08 | 0,037 | cell growth regulator with EF hand domain 1 |
| Angptl4 | 2,49 | 0,037 | angiopoietin-like 4 |

*(Continued)*

**Table 1.** (Continued)

| Differentially expressed genes in neurons | | | |
|---|---|---|---|
| Srxn1 | 2,09 | 0,037 | sulfiredoxin 1 homolog (S. cerevisiae) |
| Vim | 2,84 | 0,039 | vimentin |
| Vgf | 2,37 | 0,041 | VGF nerve growth factor inducible |
| Plpp4 | 2,33 | 0,043 | phospholipid phosphatase 4 |
| Clcf1 | 3,01 | 0,045 | cardiotrophin-like cytokine factor 1 |
| Zbtb46 | 1,58 | 0,048 | zinc finger and BTB domain containing 46 |
| Egr2 | 2,18 | 0,052 | early growth response 2 |
| Bach1 | 1,72 | 0,052 | BTB and CNC homology 1, basic leucine zipper transcription factor 1 |
| Samd4 | 1,90 | 0,052 | sterile alpha motif domain containing 4 |
| Rgs4 | 2,24 | 0,053 | regulator of G-protein signaling 4 |
| Cdkn1a | 2,86 | 0,053 | cyclin-dependent kinase inhibitor 1A (P21) |
| Adra1a | 1,92 | 0,054 | adrenergic receptor, alpha 1a |
| Csrnp1 | 2,36 | 0,054 | cysteine-serine-rich nuclear protein 1 |
| Gal | 3,64 | 0,054 | galanin and GMAP prepropeptide |
| Npas4 | 3,24 | 0,055 | neuronal PAS domain protein 4 |
| Sbno2 | 2,21 | 0,055 | strawberry notch 2 |
| Fndc9 | 3,19 | 0,061 | fibronectin type III domain containing 9 |
| Syndig1l | 1,94 | 0,063 | synapse differentiation inducing 1 like |
| Gpr3 | 1,97 | 0,075 | G-protein coupled receptor 3 |
| Fam129b | 1,40 | 0,075 | family with sequence similarity 129, member B |
| Sv2c | 2,56 | 0,075 | synaptic vesicle glycoprotein 2c |
| Adam19 | 1,62 | 0,081 | a disintegrin and metallopeptidase domain 19 (meltrin beta) |
| Pim1 | 2,43 | 0,083 | proviral integration site 1 |
| Bag3 | 1,81 | 0,083 | BCL2-associated athanogene 3 |
| Sphk1 | 2,57 | 0,086 | sphingosine kinase 1 |
| Mapkapk3 | 1,97 | 0,086 | mitogen-activated protein kinase-activated protein kinase 3 |
| Zfp36 | 2,50 | 0,086 | zinc finger protein 36 |
| Cdh4 | 1,45 | 0,087 | cadherin 4 |
| Kdm6b | 1,57 | 0,090 | KDM1 lysine (K)-specific demethylase 6B |
| Emp1 | 2,49 | 0,091 | epithelial membrane protein 1 |
| Spp1 | 3,14 | 0,091 | secreted phosphoprotein 1 |
| Sorcs3 | 2,28 | 0,094 | sortilin-related VPS10 domain containing receptor 3 |
| Cd1d2 | 3,29 | 0,097 | CD1d2 antigen |
| Prex1 | 2,06 | 0,101 | phosphatidylinositol-3,4,5-trisphosphate-dependent Rac exchange factor 1 |
| Pros1 | 2,10 | 0,101 | protein S (alpha) |
| Uck2 | 1,35 | 0,101 | uridine-cytidine kinase 2 |
| Plce1 | 1,40 | 0,101 | phospholipase C, epsilon 1 |
| Tgfb1i1 | 1,66 | 0,101 | transforming growth factor beta 1 induced transcript 1 |
| Crispld2 | 2,18 | 0,117 | cysteine-rich secretory protein LCCL domain containing 2 |
| Frrs1 | 1,87 | 0,117 | ferric-chelate reductase 1 |
| Blnk | 2,81 | 0,118 | B cell linker |
| 1700071M16Rik | 1,68 | 0,119 | RIKEN cDNA 1700071M16 gene |
| Rgs20 | 1,74 | 0,119 | regulator of G-protein signaling 20 |
| Ier2 | 2,17 | 0,120 | immediate early response 2 |
| Itprip | 1,88 | 0,130 | inositol 1,4,5-triphosphate receptor interacting protein |
| Smad7 | 1,83 | 0,130 | SMAD family member 7 |
| Svil | 1,52 | 0,130 | supervillin |
| Serinc2 | 1,79 | 0,136 | serine incorporator 2 |

*(Continued)*

**Table 1.** (Continued)

| Differentially expressed genes in neurons | | | |
|---|---|---|---|
| Cemip2 | 1,44 | 0,154 | cell migration inducing hyaluronidase 2 |
| Mir132 | 3,39 | 0,154 | microRNA 132 |
| Pear1 | 3,01 | 0,164 | platelet endothelial aggregation receptor 1 |
| Zdhhc22 | 1,85 | 0,167 | zinc finger, DHHC-type containing 22 |
| Medag | 2,23 | 0,167 | mesenteric estrogen dependent adipogenesis |
| Amotl1 | 1,71 | 0,175 | angiomotin-like 1 |
| Serpina3i | 2,75 | 0,178 | serine (or cysteine) peptidase inhibitor, clade A, member 3I |
| Ptgs1 | 2,01 | 0,178 | prostaglandin-endoperoxide synthase 1 |
| Ifit1 | 2,33 | 0,178 | interferon-induced protein with tetratricopeptide repeats 1 |
| Kcnip3 | 1,67 | 0,178 | Kv channel interacting protein 3, calsenilin |
| Odc1 | 1,57 | 0,178 | ornithine decarboxylase, structural 1 |
| Igsf9b | 2,27 | 0,178 | immunoglobulin superfamily, member 9B |
| Homer1 | 1,49 | 0,179 | homer scaffolding protein 1 |
| Spred1 | 1,62 | 0,184 | sprouty protein with EVH-1 domain 1, related sequence |
| Samd11 | 2,19 | 0,186 | sterile alpha motif domain containing 11 |
| Cdk18 | 1,98 | 0,186 | cyclin-dependent kinase 18 |
| Scd4 | 2,01 | 0,191 | stearoyl-coenzyme A desaturase 4 |
| Dgat2l6 | 3,15 | 0,191 | diacylglycerol O-acyltransferase 2-like 6 |
| Dusp4 | 1,88 | 0,191 | dual specificity phosphatase 4 |
| Anxa2 | 2,12 | 0,191 | annexin A2 |
| Serpina3n | 1,85 | 0,191 | serine (or cysteine) peptidase inhibitor, clade A, member 3N |
| Tspan9 | 1,68 | 0,191 | tetraspanin 9 |
| Eva1b | 2,00 | 0,191 | eva-1 homolog B (C. elegans) |
| Btc | 2,40 | 0,191 | betacellulin, epidermal growth factor family member |
| Acvr1c | 1,91 | 0,193 | activin A receptor, type IC |
| Rara | 1,54 | 0,194 | retinoic acid receptor, alpha |
| St8sia2 | 2,11 | 0,195 | ST8 alpha-N-acetyl-neuraminide alpha-2,8-sialyltransferase 2 |
| Tm4sf1 | 2,49 | 0,195 | transmembrane 4 superfamily member 1 |
| Cdh22 | 1,77 | 0,195 | cadherin 22 |
| Gfra1 | 1,50 | 0,202 | glial cell line derived neurotrophic factor family receptor alpha 1 |
| Itga5 | 2,11 | 0,205 | integrin alpha 5 (fibronectin receptor alpha) |
| C2cd4b | 2,11 | 0,206 | C2 calcium-dependent domain containing 4B |
| Rasa4 | 1,84 | 0,220 | RAS p21 protein activator 4 |
| Mapk6 | 1,51 | 0,223 | mitogen-activated protein kinase 6 |
| Egr4 | 1,91 | 0,227 | early growth response 4 |
| Itpkc | 1,84 | 0,227 | inositol 1,4,5-trisphosphate 3-kinase C |
| Ptx3 | 2,72 | 0,235 | pentraxin related gene |
| Tnfrsf12a | 1,87 | 0,235 | tumor necrosis factor receptor superfamily, member 12a |
| Drd1 | 1,82 | 0,246 | dopamine receptor D1 |
| Downregulated genes (N = 15) | | | |
| Gene symbol | logFC | FDR | Gene description |
| Cxcl12 | -1,96 | 0,027 | chemokine (C-X-C motif) ligand 12 |
| Ogn | -2,71 | 0,029 | osteoglycin |
| Plk5 | -2,79 | 0,040 | polo like kinase 5 |
| Cys1 | -1,85 | 0,041 | cystin 1 |
| Capn3 | -2,14 | 0,052 | calpain 3 |
| Echdc2 | -1,77 | 0,079 | enoyl Coenzyme A hydratase domain containing 2 |
| Cyp7b1 | -2,28 | 0,097 | cytochrome P450, family 7, subfamily b, polypeptide 1 |

*(Continued)*

**Table 1.** (Continued)

| Differentially expressed genes in neurons | | | |
|---|---|---|---|
| Gm12216 | -1,65 | 0,101 | predicted gene 12216 |
| Gstm6 | -1,57 | 0,161 | glutathione S-transferase, mu 6 |
| Cd34 | -1,63 | 0,167 | CD34 antigen |
| Stxbp6 | -1,51 | 0,186 | syntaxin binding protein 6 (amisyn) |
| Crlf1 | -1,97 | 0,194 | cytokine receptor-like factor 1 |
| Macrod1 | -1,52 | 0,195 | MACRO domain containing 1 |
| Gm35339 | -1,43 | 0,195 | predicted gene, 35339 |
| 6330420H09Rik | -2,15 | 0,216 | RIKEN cDNA 6330420H09 gene |

Differentially expressed genes in neurons (FDR < 0.25); logFC = log fold change; FDR = false discovery rate.

bodies and one to an upstream region of the associated gene. The other two CpGs were associated with intergenic regions.

**Differentially methylated regions.** In order to obtain information about genomic features with significantly altered differential methylation, an aggregated analysis was conducted for each genomic feature (upstream, promoter, UTR5, exon, intron, gene body, UTR3) separately. Most DMR were found in gene bodies (e.g. exonic and intronic areas) and promoters in both neurons and glia. The greatest hyper-/hypomethylated regions ratio was found at 5'UTRs (ratio 4.5:1), gene bodies (ratio 3.7:1) and promoters (ratio 3.1:1) in neurons and at promoters (ratio 8.2:1), 3'UTRs (ratio 7.0:1) and exons (ratio 5.3:1) for glia. For a full list of significantly altered DMR and their annotated GO and KEGG terms see S1 Table.

## Differential gene expression

DGE analysis compared right hippocampi of KA to SH at 24 hours post injection. After filtering, 23369 genes were used for downstream analysis. After processing, alignment and filtering, the mRNASeq samples yielded on average 7.5 giga bases (Gb) data aligned to the mouse genome (median = 9.0 Gb; range = [2.4 Gb—13.0 Gb]). The fraction aligning specifically to mRNA regions varied from 16% to 41.2% (median = 28.5), resulting in an average of 2.2 Gb per sample informative for DGE analysis (median = 2.2 Gb; range = [0.4 Gb—4.3 Gb]). In neurons, 135 genes were up- and 15 downregulated (Table 1), while in glia 147 genes were up- and 85 downregulated (Table 2). A relevant selection of broader GO / KEGG terms is presented in Tables 3 (neurons) and 4 (glia). For neuronal and glial contribution to epileptogenesis in terms of number of differentially expressed genes as part of epilepsy-relevant GO / KEGG terms, see Fig 2. For a detailed list of up- and downregulated genes and associated GO / KEGG terms see S1 Table.

**Genes up- or downregulated in both neurons and glia.** A total number of 45 genes were upregulated in both neurons and glia. GO terms of these included "positive regulation of transcription" and "cytokine mediated signaling" whilst KEGG terms contained "ECM receptor interaction", "VEGF-signaling" and "TNF-signaling". Three genes were downregulated in both neurons and glia.

## Association between differential methylation and differential gene expression

An association between DM and DGE was calculated by alignment of significantly altered DMR and differentially expressed genes (Combined FDR 0.25, for details see S1 Table and S7–S20 Figs). No general correlation between DMR and DGE at various genomic features

**Table 2. Differentially expressed genes in glia at 24 hours post injection in the intracortical kainic acid model of mTLE.**

| Differentially expressed genes in glia | | | |
| --- | --- | --- | --- |
| Upregulated genes (N = 147) | | | |
| Gene symbol | logFC | FDR | Gene description |
| Serpina3n | 4,23 | 0,001 | serine (or cysteine) peptidase inhibitor, clade A, member 3N |
| Thbd | 2,93 | 0,001 | thrombomodulin |
| Ch25h | 5,14 | 0,001 | cholesterol 25-hydroxylase |
| Lilr4b | 4,88 | 0,002 | leukocyte immunoglobulin-like receptor, subfamily B, member 4B |
| Gm3448 | 3,23 | 0,003 | predicted gene 3448 |
| Ucn2 | 8,54 | 0,003 | urocortin 2 |
| Ccl2 | 3,45 | 0,003 | chemokine (C-C motif) ligand 2 |
| Socs3 | 3,66 | 0,004 | suppressor of cytokine signaling 3 |
| Ier5l | 2,55 | 0,005 | immediate early response 5-like |
| Calca | 4,68 | 0,005 | calcitonin/calcitonin-related polypeptide, alpha |
| Sphk1 | 3,89 | 0,005 | sphingosine kinase 1 |
| Ecm1 | 2,40 | 0,005 | extracellular matrix protein 1 |
| Emp1 | 3,67 | 0,005 | epithelial membrane protein 1 |
| Ahnak2 | 3,87 | 0,005 | AHNAK nucleoprotein 2 |
| Spp1 | 4,71 | 0,005 | secreted phosphoprotein 1 |
| S1pr3 | 3,16 | 0,005 | sphingosine-1-phosphate receptor 3 |
| Fn1 | 2,64 | 0,007 | fibronectin 1 |
| Fgl2 | 2,97 | 0,008 | fibrinogen-like protein 2 |
| Timp1 | 4,60 | 0,008 | tissue inhibitor of metalloproteinase 1 |
| Tm4sf1 | 4,06 | 0,008 | transmembrane 4 superfamily member 1 |
| Rasgef1c | 3,04 | 0,009 | RasGEF domain family, member 1C |
| Ifit3 | 2,69 | 0,010 | interferon-induced protein with tetratricopeptide repeats 3 |
| Vgf | 2,83 | 0,010 | VGF nerve growth factor inducible |
| Il11 | 3,79 | 0,012 | interleukin 11 |
| Itga5 | 3,16 | 0,014 | integrin alpha 5 (fibronectin receptor alpha) |
| Iigp1 | 3,09 | 0,015 | interferon inducible GTPase 1 |
| Hmga1b | 2,26 | 0,015 | high mobility group AT-hook 1B |
| Gadd45g | 2,59 | 0,015 | growth arrest and DNA-damage-inducible 45 gamma |
| Tnc | 2,01 | 0,015 | tenascin C |
| Cd44 | 3,16 | 0,015 | CD44 antigen |
| Gpr151 | 3,26 | 0,017 | G protein-coupled receptor 151 |
| Ier3 | 2,44 | 0,020 | immediate early response 3 |
| Tnfrsf12a | 2,69 | 0,020 | tumor necrosis factor receptor superfamily, member 12a |
| Sv2c | 3,05 | 0,021 | synaptic vesicle glycoprotein 2c |
| Rasl11a | 1,99 | 0,021 | RAS-like, family 11, member A |
| Klk9 | 3,23 | 0,021 | kallikrein related-peptidase 9 |
| Btc | 3,29 | 0,024 | betacellulin, epidermal growth factor family member |
| Cebpd | 1,95 | 0,025 | CCAAT/enhancer binding protein (C/EBP), delta |
| Nptx2 | 2,91 | 0,025 | neuronal pentraxin 2 |
| Adam8 | 3,04 | 0,025 | a disintegrin and metallopeptidase domain 8 |
| Slc39a14 | 2,07 | 0,025 | solute carrier family 39 (zinc transporter), member 14 |
| Inhba | 2,61 | 0,025 | inhibin beta-A |
| Cdh22 | 2,38 | 0,025 | cadherin 22 |
| Fos | 2,95 | 0,026 | FBJ osteosarcoma oncogene |
| Rhoj | 3,00 | 0,027 | ras homolog family member J |

*(Continued)*

**Table 2.** (Continued)

| Differentially expressed genes in glia | | | |
|---|---|---|---|
| Lilrb4a | 4,02 | 0,027 | leukocyte immunoglobulin-like receptor, subfamily B, member 4A |
| Cd300lf | 3,60 | 0,028 | CD300 molecule like family member F |
| Gadd45b | 3,35 | 0,030 | growth arrest and DNA-damage-inducible 45 beta |
| Cacng5 | 2,10 | 0,030 | calcium channel, voltage-dependent, gamma subunit 5 |
| Ifi204 | 4,09 | 0,032 | interferon activated gene 204 |
| Dab2 | 2,15 | 0,036 | disabled 2, mitogen-responsive phosphoprotein |
| Myc | 2,30 | 0,036 | myelocytomatosis oncogene |
| Ifi207 | 3,11 | 0,043 | interferon activated gene 207 |
| Parp3 | 2,74 | 0,044 | poly (ADP-ribose) polymerase family, member 3 |
| Hspb1 | 3,54 | 0,046 | heat shock protein 1 |
| Trib1 | 2,04 | 0,047 | tribbles pseudokinase 1 |
| Rasip1 | 2,30 | 0,049 | Ras interacting protein 1 |
| Egr2 | 2,12 | 0,058 | early growth response 2 |
| Lpl | 1,88 | 0,058 | lipoprotein lipase |
| Tubb6 | 2,19 | 0,058 | tubulin, beta 6 class V |
| Tpbg | 1,72 | 0,058 | trophoblast glycoprotein |
| Msr1 | 3,41 | 0,058 | macrophage scavenger receptor 1 |
| Sbno2 | 2,17 | 0,058 | strawberry notch 2 |
| Gcnt2 | 2,42 | 0,061 | glucosaminyl (N-acetyl) transferase 2, I-branching enzyme |
| Fosb | 2,69 | 0,061 | FBJ osteosarcoma oncogene B |
| Serpine1 | 3,97 | 0,064 | serine (or cysteine) peptidase inhibitor, clade E, member 1 |
| Oasl2 | 2,47 | 0,064 | 2'-5' oligoadenylate synthetase-like 2 |
| Srxn1 | 1,89 | 0,065 | sulfiredoxin 1 homolog (S. cerevisiae) |
| Ptgs2 | 2,43 | 0,066 | prostaglandin-endoperoxide synthase 2 |
| Slc10a6 | 3,88 | 0,070 | solute carrier family 10 (sodium/bile acid cotransporter family), member 6 |
| Ahnak | 1,95 | 0,072 | AHNAK nucleoprotein (desmoyokin) |
| Nedd9 | 1,33 | 0,073 | neural precursor cell expressed, developmentally down-regulated gene 9 |
| Rai14 | 1,61 | 0,074 | retinoic acid induced 14 |
| Layn | 1,95 | 0,075 | layilin |
| Col16a1 | 2,51 | 0,076 | collagen, type XVI, alpha 1 |
| Atp10a | 2,07 | 0,078 | ATPase, class V, type 10A |
| Fstl4 | 1,87 | 0,078 | follistatin-like 4 |
| Wwtr1 | 1,58 | 0,080 | WW domain containing transcription regulator 1 |
| Gal | 3,44 | 0,081 | galanin and GMAP prepropeptide |
| Mx1 | 3,40 | 0,081 | MX dynamin-like GTPase 1 |
| Hmga1 | 2,18 | 0,091 | high mobility group AT-hook 1 |
| Irgm1 | 1,56 | 0,091 | immunity-related GTPase family M member 1 |
| Odc1 | 1,69 | 0,092 | ornithine decarboxylase, structural 1 |
| Gldn | 3,04 | 0,093 | gliomedin |
| Egr1 | 2,33 | 0,094 | early growth response 1 |
| Cchcr1 | 1,58 | 0,094 | coiled-coil alpha-helical rod protein 1 |
| Junb | 2,29 | 0,102 | jun B proto-oncogene |
| Slc5a3 | 1,82 | 0,102 | solute carrier family 5 (inositol transporters), member 3 |
| Socs2 | 1,76 | 0,102 | suppressor of cytokine signaling 2 |
| Il4ra | 1,81 | 0,102 | interleukin 4 receptor, alpha |
| Irf7 | 2,39 | 0,104 | interferon regulatory factor 7 |
| Nlrc5 | 2,21 | 0,104 | NLR family, CARD domain containing 5 |

(*Continued*)

**Table 2.** (Continued)

| Differentially expressed genes in glia | | | |
|---|---|---|---|
| Ptx3 | 2,99 | 0,105 | pentraxin related gene |
| Fgf18 | 2,32 | 0,107 | fibroblast growth factor 18 |
| Ifit3b | 2,41 | 0,111 | interferon-induced protein with tetratricopeptide repeats 3B |
| Strip2 | 1,74 | 0,115 | striatin interacting protein 2 |
| Has2 | 3,19 | 0,116 | hyaluronan synthase 2 |
| Mir212 | 4,52 | 0,117 | microRNA 212 |
| Flnc | 3,71 | 0,118 | filamin C, gamma |
| Map3k6 | 2,39 | 0,124 | mitogen-activated protein kinase kinase kinase 6 |
| Timeless | 1,39 | 0,124 | timeless circadian clock 1 |
| Itga7 | 1,38 | 0,132 | integrin alpha 7 |
| Bcl3 | 3,83 | 0,134 | B cell leukemia/lymphoma 3 |
| Snhg15 | 1,56 | 0,134 | small nucleolar RNA host gene 15 |
| Ccl12 | 2,59 | 0,142 | chemokine (C-C motif) ligand 12 |
| Mamstr | 2,09 | 0,142 | MEF2 activating motif and SAP domain containing transcriptional regulator |
| Clcf1 | 2,36 | 0,142 | cardiotrophin-like cytokine factor 1 |
| Bdnf | 1,81 | 0,142 | brain derived neurotrophic factor |
| Ier2 | 2,03 | 0,142 | immediate early response 2 |
| Rnf138rt1 | 5,32 | 0,149 | ring finger protein 138, retrogene 1 |
| Fosl2 | 1,59 | 0,152 | fos-like antigen 2 |
| Slfn10-ps | 2,78 | 0,160 | schlafen 10, pseudogene |
| Amotl1 | 1,65 | 0,164 | angiomotin-like 1 |
| Mir132 | 3,30 | 0,174 | microRNA 132 |
| Serpina3i | 2,64 | 0,174 | serine (or cysteine) peptidase inhibitor, clade A, member 3I |
| Hmox1 | 1,87 | 0,174 | heme oxygenase 1 |
| Lrtm2 | 1,62 | 0,175 | leucine-rich repeats and transmembrane domains 2 |
| Spred3 | 1,72 | 0,175 | sprouty-related EVH1 domain containing 3 |
| Vmn1r15 | 6,73 | 0,175 | vomeronasal 1 receptor 15 |
| Rtp4 | 1,91 | 0,177 | receptor transporter protein 4 |
| Rnf125 | 2,28 | 0,177 | ring finger protein 125 |
| Slfn2 | 2,93 | 0,182 | schlafen 2 |
| Mchr1 | 1,73 | 0,185 | melanin-concentrating hormone receptor 1 |
| Piezo2 | 1,68 | 0,185 | piezo-type mechanosensitive ion channel component 2 |
| Anxa2 | 2,01 | 0,185 | annexin A2 |
| Gpd1 | 1,68 | 0,190 | glycerol-3-phosphate dehydrogenase 1 (soluble) |
| Cyr61 | 2,08 | 0,194 | Cysteine-rich angiogenic inducer 61 |
| Plaur | 2,39 | 0,194 | plasminogen activator, urokinase receptor |
| Kdm6b | 1,32 | 0,201 | KDM1 lysine (K)-specific demethylase 6B |
| Ifit1 | 2,11 | 0,201 | interferon-induced protein with tetratricopeptide repeats 1 |
| Itga2b | 1,93 | 0,202 | integrin alpha 2b |
| Fgfr4 | 2,25 | 0,202 | fibroblast growth factor receptor 4 |
| Bst2 | 2,06 | 0,202 | bone marrow stromal cell antigen 2 |
| Gm6225 | 2,35 | 0,207 | predicted gene 6225 |
| Cbln4 | 1,60 | 0,208 | cerebellin 4 precursor protein |
| Serpina3m | 2,79 | 0,216 | serine (or cysteine) peptidase inhibitor, clade A, member 3M |
| Akap12 | 1,34 | 0,218 | A kinase (PRKA) anchor protein (gravin) 12 |
| Sdc1 | 1,59 | 0,219 | syndecan 1 |
| Ndst1 | 1,59 | 0,219 | N-deacetylase/N-sulfotransferase (heparan glucosaminyl) 1 |

(*Continued*)

**Table 2.** (Continued)

| Differentially expressed genes in glia | | | |
|---|---|---|---|
| Npas4 | 2,45 | 0,221 | neuronal PAS domain protein 4 |
| Tspan4 | 1,89 | 0,226 | tetraspanin 4 |
| Klk6 | 2,76 | 0,226 | kallikrein related-peptidase 6 |
| Cxcl10 | 2,90 | 0,226 | chemokine (C-X-C motif) ligand 10 |
| Col7a1 | 1,75 | 0,227 | collagen, type VII, alpha 1 |
| Plce1 | 1,17 | 0,237 | phospholipase C, epsilon 1 |
| Peak1 | 1,41 | 0,238 | pseudopodium-enriched atypical kinase 1 |
| Itga1 | 1,36 | 0,245 | integrin alpha 1 |
| **Downregulated genes (N = 15)** | | | |
| **Gene symbol** | **logFC** | **FDR** | **Gene description** |
| Aifm3 | -2,53 | 0,005 | apoptosis-inducing factor, mitochondrion-associated 3 |
| Sowaha | -2,14 | 0,005 | sosondowah ankyrin repeat domain family member A |
| Gdpd2 | -2,73 | 0,012 | glycerophosphodiester phosphodiesterase domain containing 2 |
| Btbd17 | -2,36 | 0,015 | BTB (POZ) domain containing 17 |
| Slc2a5 | -2,59 | 0,015 | solute carrier family 2 (facilitated glucose transporter), member 5 |
| Pcx | -2,11 | 0,016 | pyruvate carboxylase |
| Hapln1 | -2,63 | 0,025 | hyaluronan and proteoglycan link protein 1 |
| Shroom2 | -2,28 | 0,026 | shroom family member 2 |
| Gpr12 | -2,22 | 0,039 | G-protein coupled receptor 12 |
| Ccdc13 | -1,80 | 0,046 | coiled-coil domain containing 13 |
| Fn3k | -2,03 | 0,049 | fructosamine 3 kinase |
| P2ry12 | -2,57 | 0,053 | purinergic receptor P2Y, G-protein coupled 12 |
| Cygb | -1,88 | 0,053 | cytoglobin |
| Ankub1 | -2,23 | 0,060 | ankrin repeat and ubiquitin domain containing 1 |
| Siglech | -2,16 | 0,061 | sialic acid binding Ig-like lectin H |
| Itpka | -1,70 | 0,061 | inositol 1,4,5-trisphosphate 3-kinase A |
| Traf4 | -1,81 | 0,065 | TNF receptor associated factor 4 |
| Hpca | -1,84 | 0,074 | hippocalcin |
| Ppp1r1b | -1,75 | 0,077 | protein phosphatase 1, regulatory inhibitor subunit 1B |
| Nkain4 | -2,55 | 0,077 | Na+/K+ transporting ATPase interacting 4 |
| Folh1 | -2,24 | 0,077 | folate hydrolase 1 |
| Kctd4 | -2,09 | 0,081 | potassium channel tetramerisation domain containing 4 |
| Gstm6 | -1,67 | 0,083 | glutathione S-transferase, mu 6 |
| Shisa8 | -2,21 | 0,091 | shisa family member 8 |
| 2810468N07Rik | -2,22 | 0,092 | RIKEN cDNA 2810468N07 gene |
| Abca9 | -1,97 | 0,092 | ATP-binding cassette, sub-family A (ABC1), member 9 |
| Paqr7 | -1,93 | 0,099 | progestin and adipoQ receptor family member VII |
| Chn1 | -1,71 | 0,099 | chimerin 1 |
| Ntsr2 | -2,14 | 0,105 | neurotensin receptor 2 |
| Myh14 | -1,76 | 0,105 | myosin, heavy polypeptide 14 |
| Nwd1 | -1,82 | 0,108 | NACHT and WD repeat domain containing 1 |
| Fam234a | -1,77 | 0,109 | family with sequence similarity 234, member A |
| Susd5 | -1,88 | 0,116 | sushi domain containing 5 |
| Faah | -1,50 | 0,117 | fatty acid amide hydrolase |
| Tppp3 | -1,76 | 0,124 | tubulin polymerization-promoting protein family member 3 |
| Abca6 | -1,46 | 0,124 | ATP-binding cassette, sub-family A (ABC1), member 6 |
| Gnai1 | -1,90 | 0,134 | guanine nucleotide binding protein (G protein), alpha inhibiting 1 |

*(Continued)*

**Table 2.** (Continued)

| Differentially expressed genes in glia | | | |
|---|---|---|---|
| Cfap100 | -1,48 | 0,135 | cilia and flagella associated protein 100 |
| Grm3 | -2,01 | 0,142 | glutamate receptor, metabotropic 3 |
| Phgdh | -1,66 | 0,149 | 3-phosphoglycerate dehydrogenase |
| Selplg | -2,14 | 0,152 | selectin, platelet (p-selectin) ligand |
| Epn2 | -1,61 | 0,171 | epsin 2 |
| 2900052N01Rik | -2,06 | 0,174 | RIKEN cDNA 2900052N01 gene |
| Rlbp1 | -1,78 | 0,175 | retinaldehyde binding protein 1 |
| Pantr1 | -1,72 | 0,175 | POU domain, class 3, transcription factor 3 adjacent noncoding transcript 1 |
| Nat8f4 | -1,42 | 0,175 | N-acetyltransferase 8 (GCN5-related) family member 4 |
| Plk5 | -2,14 | 0,175 | polo like kinase 5 |
| Nat8f1 | -1,91 | 0,175 | N-acetyltransferase 8 (GCN5-related) family member 1 |
| 1700066M21Rik | -1,65 | 0,175 | RIKEN cDNA 1700066M21 gene |
| Adi1 | -1,61 | 0,176 | acireductone dioxygenase 1 |
| Tmem191c | -1,45 | 0,177 | transmembrane protein 191C |
| Gmnc | -2,55 | 0,177 | geminin coiled-coil domain containing |
| Zfp763 | -1,51 | 0,182 | zinc finger protein 763 |
| Slc25a18 | -1,79 | 0,185 | solute carrier family 25 (mitochondrial carrier), member 18 |
| Hhip | -2,01 | 0,185 | Hedgehog-interacting protein |
| Calb1 | -1,51 | 0,185 | calbindin 1 |
| Chst5 | -1,74 | 0,185 | carbohydrate (N-acetylglucosamine 6-O) sulfotransferase 5 |
| Trim59 | -2,18 | 0,185 | tripartite motif-containing 59 |
| Gpr34 | -2,22 | 0,185 | G protein-coupled receptor 34 |
| Olfml1 | -2,24 | 0,185 | olfactomedin-like 1 |
| Mturn | -1,41 | 0,185 | maturin, neural progenitor differentiation regulator homolog (Xenopus) |
| Gstm1 | -1,80 | 0,185 | glutathione S-transferase, mu 1 |
| Enho | -1,63 | 0,185 | energy homeostasis associated |
| Prodh | -1,86 | 0,185 | proline dehydrogenase |
| Slc27a1 | -1,71 | 0,185 | solute carrier family 27 (fatty acid transporter), member 1 |
| Pacsin3 | -1,44 | 0,185 | protein kinase C and casein kinase substrate in neurons 3 |
| Htr1a | -1,95 | 0,185 | 5-hydroxytryptamine (serotonin) receptor 1A |
| Dll3 | -1,72 | 0,187 | delta like canonical Notch ligand 3 |
| Map6d1 | -1,60 | 0,187 | MAP6 domain containing 1 |
| Prrg1 | -1,61 | 0,193 | proline rich Gla (G-carboxyglutamic acid) 1 |
| Carns1 | -1,88 | 0,201 | carnosine synthase 1 |
| Tle2 | -1,48 | 0,201 | transducin-like enhancer of split 2 |
| Macrod1 | -1,45 | 0,202 | MACRO domain containing 1 |
| Nrgn | -1,51 | 0,204 | neurogranin |
| Plin3 | -2,18 | 0,207 | perilipin 3 |
| Grhpr | -1,38 | 0,208 | glyoxylate reductase/hydroxypyruvate reductase |
| Sult1a1 | -2,19 | 0,214 | sulfotransferase family 1A, phenol-preferring, member 1 |
| Pls1 | -1,58 | 0,216 | plastin 1 (I-isoform) |
| Lin7b | -1,69 | 0,216 | lin-7 homolog B (C. elegans) |
| Armh4 | -1,53 | 0,218 | armadillo-like helical domain containing 4 |
| Panx2 | -1,33 | 0,226 | pannexin 2 |
| Appl2 | -1,76 | 0,227 | adaptor protein, phosphotyrosine interaction, PH domain and leucine zipper containing 2 |
| Grhl1 | -1,01 | 0,227 | grainyhead like transcription factor 1 |
| Tmem255b | -1,65 | 0,233 | transmembrane protein 255B |

(*Continued*)

**Table 2.** (Continued)

| Differentially expressed genes in glia | | | |
|---|---|---|---|
| Pigz | -1,71 | 0,243 | phosphatidylinositol glycan anchor biosynthesis, class Z |

Differentially expressed genes in glia (FDR < 0.25); logFC = log fold change; FDR = false discovery rate.

(upstream, promoter, UTR5, exon, intron, UTR3) was found (S1 Table and S7–S20 Figs). However, significant DMR coincided with DGE at 41 loci for neurons and 12 loci for glia (Tables 5 and 6).

## Discussion

Neurons and glial cells execute specific complementary tasks in normal brain functioning as well as in the pathological processes precipitating neurological diseases [9, 48, 49]. This diversity is represented on the transcriptome- [36, 50] and epigenome level [34, 35]. In this study we investigated neuronal and glial alterations of DNAm and gene expression and their possible association in a mouse model of epilepsy. In order to explore the epigenetic and transcriptomic signature of both cell types in early epileptogenesis, we separated neurons and glia by FANS [51]. This approach was recently applied in epigenetic studies [39–41]. We identified specific neuronal and glial DNAm and DGE changes at particular genomic loci, potentially including important upstream mechanisms worth further investigation.

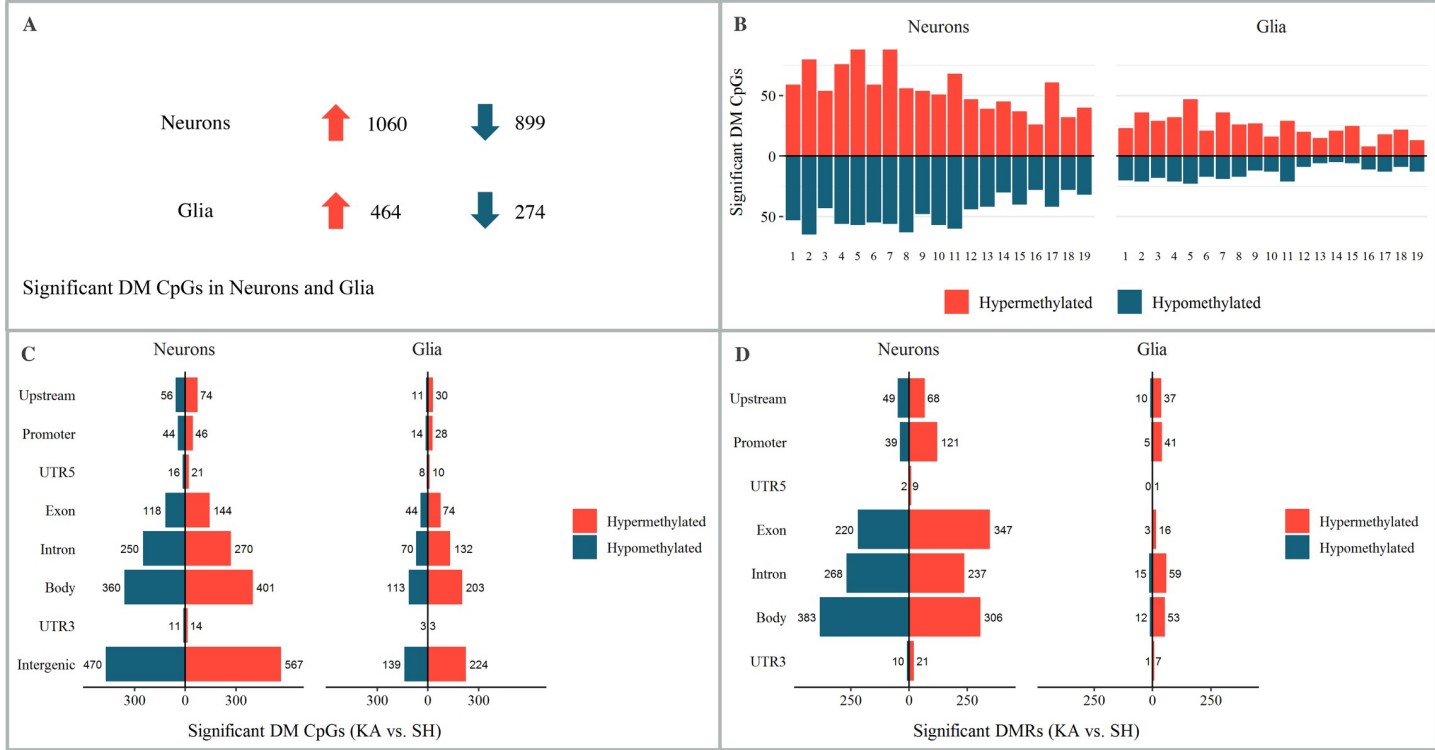

**Fig 1. Differential DNA methylation at 24 hours post injection in the intracortical kainic acid model of mTLE.** (A)–(C) Statistical analysis of differentially methylated CpGs (DM CpGs) of KA versus SH group, 24 hours post injection. (A) DM CpGs in neurons and glia; upward arrow indicates hypermethylation and downward arrow hypomethylation of DM CpGs. (B) Chromosomal distribution of DM CpGs. (C) Distribution of DM CpGs amongst genomic features. (D) Distribution of differentially methylated regions (DMR) amongst genomic features.

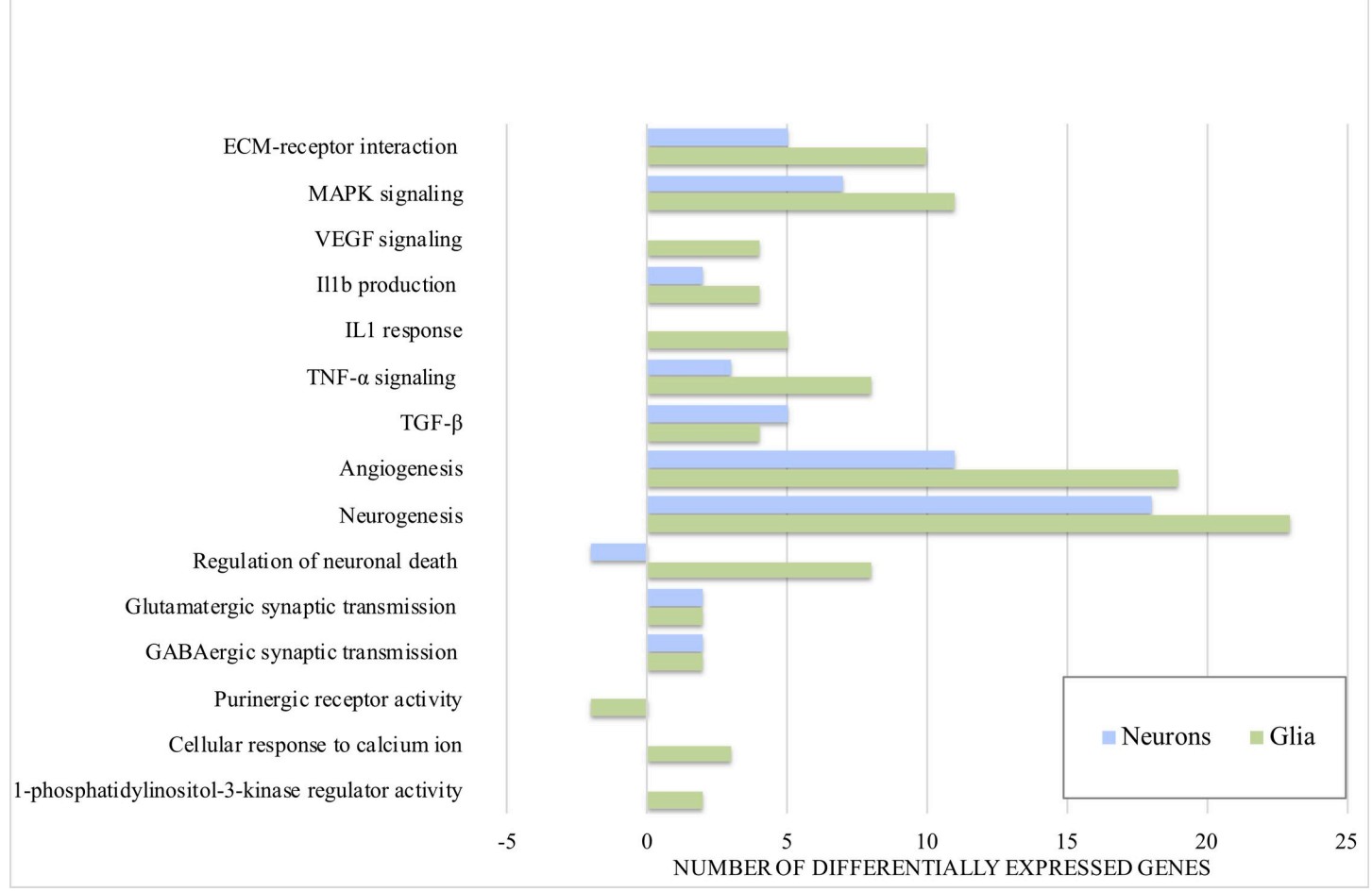

**Fig 2. Number of up- and downregulated neuronal and glial genes within epilepsy-relevant functional annotation terms (GO / KEGG).** Neuronal and glial contribution to epilepsy-relevant GO / KEGG pathways (p<0.05) amongst differentially expressed genes. Negative values indicate downregulated genes, positive values upregulated genes.

### Differential methylation

With an overlap of only 0.22% of DM CpGs between neurons and glia, differential methylation occurs primarily in a cell specific manner during early epileptogenesis. Accordingly, the attributed GO terms to DM CpGs and DMR reveal mostly neuronal and glia specific terms (S1 Table). Apart from a near even ratio at neuronal promoters, we observe an overweight of hypermethylated to hypomethylated CpG sites at 24 hours post injection. In early stages of epileptogenesis, previous studies suggest no general alteration of DNAm [30] or a tendency towards hypomethylation [52]. In chronic phases of epilepsy, both hyper- and hypomethylation have been reported [29, 53]. Differences in pro-convulsant agents, mode and region at which they are applicated, stage of epileptogenesis, anatomical regions investigated, as well as the number of CpG sites covered, may account for varying results. A comparison of genes associated with significant DM CpGs and DMR from our study with results from hippocampal tissue from chronic TLE patients [54] unvails marginal but interesting overlaps. This may be due to different methylation analysis methods and stage specific character of hippocampal DNAm during epileptogenesis. A comparison of our results to those from peripheral blood of TLE patients [55], reveals 15 overlapping DM CpGs and 11 overlapping DMRs.

**Table 3. Relevant GO and KEGG terms of up- and downregulated genes in neurons at 24 hours post injection in the intracortical kainic acid model of mTLE.**

|  | Upregulated genes in neurons (N = 135) | Downregulated genes in neurons (N = 15) |
|---|---|---|
| GO* | • Cell differentiation | • Lymphocyte migration |
|  | Signal transduction | • Leukocyte migration |
|  | • Cell death | • Endothelial cell proliferation |
|  | • Regulation of gene expression | • Regeneration |
|  | • Cell-cell signaling | • Cellular response to DNA damage stimulus |
|  | • Cell surface receptor signaling | • Growth factor activity |
|  | • Cell growth | • Vesicle organization |
| KEGG* | • ECM-receptor interaction |  |
|  | • MAPK signaling |  |
|  | • IL17 signaling |  |
|  | • cAMP signaling |  |
|  | • TNF signaling |  |
|  | • VEGF signaling |  |
|  | • TGFbeta signaling |  |

Relevant GO and KEGG terms associated with significantly (FDR 0.25) upregulated genes

* = $p < 0.05$.

## Differential gene expression

Most differentially expressed genes detected in this study, were specific to either neurons or glia (S1 Table). Only 45 genes were commonly up- and three downregulated in neurons and glia. At this early stage of epileptogenesis, glial cells apparently contribute a higher number of altered gene transcripts than neurons within epilepsy-related GO and KEGG terms (Fig 2). Many significantly differentially expressed genes found in this epilepsy model overlap with data from other experimental and human TLE studies [29, 56–58]. Comparing our differentially expressed genes in neurons and glia with differential gene expression from various epilepsy models and stages of TLE [58], glia contributes with more upregulated gene transcripts (13 glia and 11 in neurons), supporting the notion that glia contributes essentially to epileptogenesis [59–66]. Neurons contribute with slightly higher number of altered gene transcripts when comparing to a amygdala stimulation model of epilepsy while glia contributes with a higher number of differentially expressed genes when comparing to a traumatic brain injury model of epilepsy [29]. Within up- and downregulated gene transcripts from a study on refractory human TLE, neurons and glia exhibit an almost even number of gene transcripts [57]. For an overview of essential up- and downregulated pathways see Tables 1 and 2. A summary of neuronal and glial contribution (number of genes within a GO / KEGG term) to epileptogenesis is presented in Fig 2.

## Altered epilepsy-relevant pathways based on differentially expressed genes

*Upregulation of growth arrest and DNA-damage-inducible beta/gamma* (GADD45B/G). One of the genes upregulated in both neurons and glia is *GADD45G*, a member of the environmental stress inducible *GADD45*-like genes that mediate activation of various pathways, including c-Jun N-terminal protein kinase family of mitogen-activated protein kinases [67]. It has previously been shown to be elevated after KA induced status epilepticus [68] and to possess DNA demethylation qualities [69], thus potentially linking epileptic activity to changes in DNA methylation. We also find elevated mRNA levels of *GADD45B*, which is another member of

**Table 4. Relevant GO and KEGG terms of up- and downregulated genes in glia at 24 hours post injection in the intracortical kainic acid model of mTLE.**

|  |  | Upregulated genes in glia (N = 147) | Downregulated genes in glia (N = 85) |
|---|---|---|---|
| GO* |  | • Regulation of IL1 | • Purinergic nucleotide receptor activity |
|  |  | • Response to IL1 | • Regulation of ion transport |
|  |  | • MAPK | • PLC activating G-protein receptor |
|  |  | • Apoptotic process | • NAD binding |
|  |  | • Lymphocyte chemotaxis | • Glutamate receptor signaling |
|  |  | • Cytokine production | • Myelin sheath |
|  |  | • Angiogenesis | • Mitochondrial part |
| KEGG* |  | • ECM-receptor interaction | • Glutathione metabolism |
|  |  | • MAPK signaling | • Pyruvate metabolism |
|  |  | • PI3K-Akt signaling pathway | • ABC transporters |
|  |  | • Cytokine-cytokine receptor interaction | • cAMP signaling pathway |
|  |  | • TNF signaling | • Glycine, serine and threonine metabolism |
|  |  | • VEGF signaling |  |
|  |  | • JAK-stat signaling |  |

Relevant GO and KEGG terms associated with significantly (FDR 0.25) upregulated genes
* = p<0.05.

the *GADD45*-like genes. GADD45B has recently been shown to promote neuronal activity induced neurogenesis via demethylation of the BDNF and fibroblast growth factor promoter, linking neuronal activity to DNA methylation alterations [69].

## Upregulation of sphingosine-kinase 1 (SPHK1) and sphingosine 1 receptor 3 (S1R3)

Another interesting finding is the upregulation of *SPHK1* mRNA in neurons and glia. SPHK1 phosphorylates sphingosine to sphingosine-1-phosphate (S1P) [70]. S1P in turn is involved in neural development, signaling, autophagy and neuroinflammation as well as a plethora of pathological central nervous conditions [71] and has been shown to modulate histone deacetylase activity [72]. A recent study revealed antiepileptogenic effects of fingolimod [71], a SP1-receptor modulator and FDA approved drug for the treatment of multiple sclerosis [73], possibly via attenuation of astro- [74] and microglial [75] reactions. Further, we find elevated expression of *S1R3*, a S1P receptor, in glia. This receptor has been shown to be elevated in hippocampi of kainic acid and pilocarpine epilepsy models as well as in humans with TLE, and is mainly expressed in astrocytes [76].

## Upregulated mitogen-activated protein kinase (MAPK) pathways in neurons and glia

MAPK is a type of protein kinase specific to the amino acids serine and threonine. MAPKs are involved in directing cellular responses to a variety of different stimuli, such as proinflammatory cytokines, mitogens, osmotic stress and heat shock. They regulate cell functions including proliferation, gene expression and differentiation, mitosis, cell survival and apoptosis [77]. We find several genes within MAPK pathways (GO / KEGG) to be expressed more in the KA than SH group with glia contributing a higher number of differentially expressed genes within the pathways than neurons (S1 Table). In the context of epilepsy, MAPK are thought to play a role in Cx43 phosphorylation, involving TNF-α, interleukin (IL)-1b [78] and VEGF [79]. We find

**Table 5. Association between DMR and DGE in neurons at 24 hours post injection.**

| Genomic feature | Neurons | | | |
| | Gene symbol | DM logFC | DGE logFC | Gene description |
| --- | --- | --- | --- | --- |
| Upstream | ZBTB18 | 3,22 | -1,20 | Zinc finger and BTB domain-containing protein 18 |
| | SPP1 | -0,99 | 3,14 | Osteopontin |
| Promoter | SPP1 | -0,97 | 3,14 | Osteopontin |
| | PDE6B | -2,72 | 2,91 | Rod cGMP-specific 3',5'-cyclic phosphodiesterase subunit beta |
| | BTG2 | 5,23 | 1,49 | Protein BTG2 |
| | DPYSL3 | 4,70 | 1,02 | Dihydropyrimidinase-related protein 3 |
| | GADD45G | 4,59 | 2,54 | Growth arrest and DNA damage-inducible protein GADD45 gamma |
| | SCRT2 | 3,77 | 1,43 | Transcriptional repressor scratch 2 |
| Exon | TRIB2 | 4,72 | 1,20 | Tribbles homolog 2 |
| | ZFP36 | 1,77 | 2,50 | mRNA decay activator protein ZFP36 |
| | GAL | 2,13 | 3,64 | Galanin |
| | CXCL12 | -0,87 | -1,96 | Stromal cell-derived factor 1 |
| | SV2C | -1,27 | 2,56 | Synaptic vesicle glycoprotein 2C |
| | GM5577 | 3,67 | -1,36 | Predicted gene 5577 |
| | LTBP1 | -1,11 | 1,48 | Latent transforming growth factor beta binding protein 1 |
| | ARL4D | -0,61 | 1,61 | ADP-ribosylation factor-like 4D |
| | FNDC9 | -1,07 | 3,19 | Fibronectin type III domain-containing protein 9 |
| | CD34 | 3,18 | -1,63 | Hematopoietic progenitor cell antigen CD34 |
| | EPHX1 | 1,99 | -1,28 | Epoxide hydrolase 1 |
| | TUBB6 | -0,51 | 1,49 | Tubulin beta-6 chain |
| | BACH1 | -1,04 | 1,72 | Fanconi anemia group J protein homolog |
| Intron | SAMD11 | 0,73 | 2,19 | sterile alpha motif domain containing 11 |
| | KIF18A | 0,91 | 1,46 | Kinesin-like protein KIF18A |
| | NPTX2 | -0,69 | 3,61 | Neuronal pentraxin-2 |
| | IGF2BP2 | -0,56 | 2,04 | Insulin-like growth factor 2 mRNA-binding protein 2 |
| | NPTX2 | -0,69 | 3,61 | Neuronal pentraxin-2 |
| Gene body | KIF18A | 0,80 | 1,46 | Kinesin family member 18A |
| | GAL | 2,09 | 3,64 | Galanin peptides |
| | TRIB2 | 2,07 | 1,20 | tribbles pseudokinase 2 |
| | SAMD11 | 0,70 | 2,19 | Sterile alpha motif domain-containing protein 11 |
| | ADGRF4 | 0,92 | 1,66 | Adhesion G protein-coupled receptor F4 |
| | LTBP1 | -0,52 | 1,48 | Latent-transforming growth factor beta-binding protein 1 |
| | FAM129B | -0,59 | 1,40 | Niban-like protein 1 |
| | IGF2BP2 | -0,54 | 2,04 | insulin-like growth factor 2 mRNA binding protein 2 |
| | CD34 | 3,30 | -1,63 | CD34 antigen |
| | SORCS3 | -0,40 | 2,28 | VPS10 domain-containing receptor SorCS3 |
| | GXYLT2 | -0,85 | 1,40 | Glucoside xylosyltransferase 2 |
| | SV2C | -0,63 | 2,56 | Synaptic vesicle glycoprotein 2C |
| | ARL4D | -0,60 | 1,61 | ADP-ribosylation factor-like protein 4D |
| UTR3 | EGR3 | 3,45 | 1,61 | Early growth response protein 3 |
| | NEDD9 | -1,46 | 1,51 | Enhancer of filamentation 1 |

Genetic loci with coincidence of significant DM and DGE (FDR 0.25); logFC = log fold change.

elevated expression levels of genes within KEGG pathways for both TNF-α (mostly in glia) and VEGF (slightly more in neurons, see S1 Table). Phosphorylation of Cx43 in turn has been

**Table 6. Association between DMR and DGE in glia at 24 hours post injection.**

| Genomic feature | Glia | | | |
| | Gene symbol | DM logFC | DGE logFC | Gene description |
|---|---|---|---|---|
| Upstream | ZBTB46 | -1,01 | 0,87 | Zinc finger and BTB domain-containing protein 46 |
| | KIRREL2 | 3,75 | -1,41 | Kin of IRRE-like protein 2 |
| | PANTR1 | -4,55 | -1,72 | POU3F3 Adjacent Non-Coding Transcript 1 (non coding RNA) |
| | ASCL1 | -4,05 | -1,42 | Achaete-scute homolog 1 |
| Promoter | HDAC11 | 4,28 | -1,42 | Histone deacetylase 11 |
| | DRD1 | 4,24 | 1,62 | D(1A) dopamine receptor |
| Exon | ZFP467 | 0,47 | -1,38 | Zinc finger protein 467 |
| Intron | DRD1 | 4,17 | 1,62 | D(1A) dopamine receptor |
| | 2810468N07RIK | -3,95 | -2,22 | RIKEN cDNA (lncRNA) |
| Gene body | - | | | |
| UTR3 | FGFBP3 | 3,75 | -1,07 | Fibroblast growth factor-binding protein 3 |

Genetic loci with coincidence of significant DM and DGE (FDR 0.25); logFC = log fold change.

associated with its elevated internalization and degradation [80], possible contributing to astrocyte uncoupling in both mTLE mice and humans [15].

*Astrocytic calcium signaling pathways altered in glia.* Neuronal activity induced elevations in astrocytic intracellular calcium levels may in turn facilitate astrocytic release of neuroactive substances including glutamate, aggravating epileptic activity [81]. In acute stages of epileptogenesis, calcium transients in astrocytes are increased, possibly contributing to elevated extracellular potassium levels via Calcium-dependent protease induced cleavage of the dystrophin associated protein complex [82, 83]. Elevated extracellular potassium levels in turn may lead to increased excitability of neurons and thereby generate epileptiform activity [84]. We also find elevated gene expression of inositol 1,4,5-trisphosphate 3-kinase A (*ITPKA*), a protein kinase inactivating inositol triphosphate dependent calcium release from the astrocyte endoplasmic reticulum [85, 86], in glia. Further, we find 1-Phosphatidylinositol-4,5-bisphosphate phosphodiesterase epsilon-1 (*PLCE1*), a member of the phosphatidylinositol-specific phospholipase C family that via G-protein coupled receptors are involved in Inositol-triphosphate and diacylglycerol generation and as such mediate intracellular Calcium elevation [87], elevated in glia. Thirdly, we find *CACNG5*, a calcium permissive AMPA receptor subunit [88], elevated in glia. Possibly all three genes mediate pro-epileptogenic effects via astrocytic calcium signaling. For a complete summary of differentially expressed genes see S1 Table.

## Relationship between differential methylation and differential gene expression

We did not find a general correlation of DM and DGE at specific genomic regions (S1 Table, S7–S20 Figs). However, DM coincided with DGE at 41 genes for neurons and 10 genes for glia. Our results are in line with previous studies that did not find a general correlation between DNA methylation and gene expression in epilepsy, but rather a number of singular genes where significant DM with DGE coincided [30, 89, 90]. Other studies did report a certain degree of general association between DM at specific genomic regions and DGE [29]. For a full list of coinciding alterations in DM and DGE see Table 5. The identified coinciding alterations of DM and DGE in this study are relatively few and their role in epileptogenesis remains uncertain. Interestingly, they point to genes and pathways previously implicated in epilepsy, TLE and epileptogenesis. In the following we present a selection of these in depth.

**Coinciding alterations of differential methylation and differential gene expression in neurons.** *Osteopontin* (SPP1) *promoter hypomethylation associated with elevated gene expression.* SPP1 mediates diverse aspects of cellular functioning in the central nervous system, e.g. the recruitment and activation of microglia and astrocytes, the cumulative effect possibly being neuroprotective [91]. In multiple sclerosis, SPP1 mediates pro-inflammatory pathways contributing to the relapse remission phenotype via e.g. NF-κB [92]. In our study, *SPP1* mRNA is significantly upregulated in both neurons and glia, and in neurons this elevated gene expression is associated with significant hypomethylation at the associated upstream region and promoter. These results confirm previous findings of elevated *SPP1* mRNA levels in epilepsy [93]. We further found elevated mRNA levels of *CD44*, a Osteopontin receptor [94] involved in epileptogenesis [95], in glia.

*Hypermethylation at a Galanin* (GAL) *exon associated with elevated gene expression.* *GAL* encodes the neuropeptide Galanin which previously has been shown to possess seizure attenuating properties and discussed as a possible antiepileptogenic target [96]. Further, in a recent study, a de novo mutation in GAL has been unveiled as a possible cause for TLE [97].

In our study, *GAL* mRNA is significantly upregulated in both glia and neurons (slightly higher log fold change (logFC) and lower FDR in neurons) and we find a significant association of hypermethylation of an exonic region of *GAL* with elevated gene expression in neurons. Thus, the hypermethylation at the exonic region of *GAL* with possible consecutive elevated levels of GAL might represent a crucial endogenic seizure attenuating mechanism.

## *Hypomethylation at synaptic vesicle protein 2 c* (SV2C) *exon associated with upregulated gene expression*

SV2C is, together with SV2A and SV2B, part of the family of synaptic vesicle proteins that are involved in $Ca^{2+}$ dependent synaptic vesicle exocytosis and neurotransmission [98]. The most prominent epilepsy-related member, SV2A, is the main target through which levetiracetam and brivaracetam exert their antiepileptic and possibly antiepileptogenic effects [99]. In hippocampi of patients with chronic TLE, SV2C was the only of three synaptic vesicle proteins found to be significantly elevated. It was associated with mossy fiber sprouting and glutamatergic synapses and was proposed as a potential antiepileptogenic target [100]. Recent findings suggest a role of SV2C in the disruption of dopamine signaling in Parkinson's Disease [101]. At 24 hours post injection, we find elevated levels of *SV2C* mRNA in glia and neurons. In neurons this elevated gene expression is associated with significant hypomethylation of its exonic regions. Hypomethylation of the *SV2C* exon may thus exert upstream pro-epileptogenic effects.

## Coinciding alterations of differential methylation and differential gene expression in glia

*Promoter hypermethylation at* HDAC11 *associated with reduced gene expression levels.* In line with previous results [102], we find *HDAC11* mRNA levels decreased after SE. This reduced gene expression coincides with a significantly increased methylation at its associated promoter in glia. Reduced levels of *HDAC11* may cause an increased acetylation at H4 [103], previously shown to correlate with elevated levels of c-fos, c-jun and BDNF [104]. BDNF in turn has been associated with seizure-aggravating effects in acute phases of epileptogenesis [105] and higher levels of the microRNA miR-132, possibly via ERK and MAPK pathways [106]. Further, miR-132 has recently been associated with seizure induced neuronal apoptosis [107]. We find elevated expression levels of *BDNF* (only glia), miR-132 (both neurons and glia) and MAPK-(glia more transcripts than neurons) and ERK- (glia more than neurons) pathways in early

epileptogenesis. The hypermethylation of the *HDAC11* promoter with its possible downstream effects ultimately leading to elevated levels of *BDNF* and miR-132 might represent a possible antiepileptogenic target for site specific alteration of DNAm.

**Hypermethylation at the intron and promoter at dopamine receptor D1 (DRD1) associated with elevated DRD1 mRNA levels.** Dopamine exerts its seizure inducing effects via DRD1 mediated ERK1/2 pathways [108]. We find elevated levels of *DRD1* mRNA in both neurons and glia. In glia, this augmentation in gene expression is associated with significant hypermethylation in the intronic region and hypermethylation at the promoter of *DRD1*. Hypermethylation at glial *DRD1* (intronic region) may facilitate epileptogenesis.

## Technical limitations

This study features several technical limitations worth mentioning. Firstly, we cannot rule out that the NeuN- fraction contains a minor number of non-glial cells (pericytes, endothelial cells) [109, 110]. RRBS associated technical limitations involve loss of information associated with e.g. msp1 enzyme cleavage, library pooling/ fragment size selection, bisulfite conversion, sequencing (depth/coverage) [43]. The use of nuclear mRNA results in enrichment of mRNA coding for proteins with nuclear functions [111] and a potentially lower level of immediate early genes [112]compared to when using cytosolic mRNA [113].

## Conclusion

In this study, we found DNAm and DGE in early epileptogenesis to occur primarily in a cell-specific manner. We identified several potential neuronal and glial upstream targets worth further investigation. Information on the cellular origin of epigenomic and transcriptomic effects increases our understanding of involved pathological processes and provides a basis for possible future cell specific therapeutic approaches.

## Supporting information

**S1 Table. DM and DGE in neurons and glia.** Table of significantly altered mRNA (RRBS) and DNA methylation (DM CpGs and DMR) in neurons and glia as well as overviews and adjunct functional annotations (GO / KEGG).
(XLSX)

**S1 Supporting Information. Detailed methods.**
(DOCX)

**S1 Fig. Flowchart of tissue processing from hippocampi to NeuN+ / NeuN- nuclei.** Hippocampi from Kainate (n = 8) or Sham (n = 8) animals at 24 hrs. after injection were pooled (sample 1: pooled 4 to 1; sample 2 and 3: pooled 2 to 1) and homogenized to obtain single nuclei. The nuclei were filtered, centrifugated, pelleted and resuspended, before being subjected to FANS.
(TIF)

**S2 Fig. Sorting of NeuN-positive and NeuN-negative nuclei by flow cytometry (A–F) Nuclei were defined as PI-positive events, and aggregated nuclei were excluded in an SSC-w vs FSC-a plot.** Single nuclei from a tissue not expressing NeuN (adult mouse liver) were used to define the NeuN-positive and NeuN-negative gates (A), and hippocampal nuclei were sorted accordingly (B).
(TIF)

**S3 Fig. Estimated bisulfite conversion rates.** The left panel shows the PCT_NON_CPG_BA-SES_CONVERTED metric computed by Picard/CollectRRBSMetrics. This is defined as the fraction of converted cytosines among all non-CpG cytosines encountered in the sequencing data. The right panel shows the observed conversion rate of the unmethylated "end-repair" cytosines added in the RRBS prep (see methods for details).
(TIF)

**S4 Fig. Principal component analysis (MDS) of RRBS-Data.** The principal component analysis of RRBS data distinguishes clearly between neurons and glia but not between KA and SH.
(TIF)

**S5 Fig. Principal component analysis (MDS) of RNA-Data.** Principal component analysis of mRNAseq data clearly distinguished between neurons and glia as well as KA and SH.
(TIF)

**S6 Fig. Expression levels (mRNAseq, normalized counts) for CNS cell type specific genes.** Neurons: *RBFOX3* (= NeuN), Astrocytes: *ALDH1L1*, Microglia: *CX3CR1*, Oligodendrocytes: *MBP*, Pericytes: *PDGFRB*, Endothelial cells: *PECAM1*; Expression in the NeuN+ and NeuN-fraction on the left and right side of each graph.
(TIF)

**S7 Fig. DM and DGE (neurons, upstream).** Visualization of DM and DGE (FDR 0.25) for neurons (upstream). Genes associated with significantly altered DMR and DGE are indicated in the figure.
(TIF)

**S8 Fig. DM and DGE (glia, upstream).** Visualization of DM and DGE (FDR 0.25) for glia (upstream). Genes associated with significantly altered DMR and DGE are indicated in the figure.
(TIF)

**S9 Fig. DM and DGE (neurons, promoter).** Visualization of DM and DGE (FDR 0.25) for neurons (promoters). Genes associated with significantly altered DMR and DGE are indicated in the figure.
(TIF)

**S10 Fig. DM and DGE (glia, promoter).** Visualization of DM and DGE (FDR 0.25) for glia (promoters). Genes associated with significantly altered DMR and DGE are indicated in the figure.
(TIF)

**S11 Fig. DM and DGE (neurons, UTR5).** Visualization of DM and DGE (FDR 0.25) for neurons (UTR5). Genes associated with significantly altered DMR and DGE are indicated in the figure.
(TIF)

**S12 Fig. DM and DGE (glia, UTR5).** Visualization of DM and DGE (FDR 0.25) for glia (UTR5). Genes associated with significantly altered DMR and DGE are indicated in the figure.
(TIF)

**S13 Fig. DM and DGE (neurons, exon).** Visualization of DM and DGE (FDR 0.25) for neurons (exon). Genes associated with significantly altered DMR and DGE are indicated in the figure.
(TIF)

**S14 Fig. DM and DGE (glia, exon).** Visualization of DM and DGE (FDR 0.25) for glia (exon). Genes associated with significantly altered DMR and DGE are indicated in the figure. (TIF)

**S15 Fig. DM and DGE (neurons, intron).** Visualization of DM and DGE (FDR 0.25) for neurons (intron). Genes associated with significantly altered DMR and DGE are indicated in the figure. (TIF)

**S16 Fig. DM and DGE (glia, intron).** Visualization of DM and DGE (FDR 0.25) for glia (intron). Genes associated with significantly altered DMR and DGE are indicated in the figure. (TIF)

**S17 Fig. DM and DGE (neurons, gene body).** Visualization of DM and DGE (FDR 0.25) for neurons (gene body). Genes associated with significantly altered DMR and DGE are indicated in the figure. (TIF)

**S18 Fig. DM and DGE (glia, gene body).** Visualization of DM and DGE (FDR 0.25) for glia (gene body). Genes associated with significantly altered DMR and DGE are indicated in the figure. (TIF)

**S19 Fig. DM and DGE (neurons, UTR3).** Visualization of DM and DGE (FDR 0.25) for neurons (UTR3). Genes associated with significantly altered DMR and DGE are indicated in the figure. (TIF)

**S20 Fig. DM and DGE (glia, UTR3).** Visualization of DM and DGE (FDR 0.25) for glia (UTR3). Genes associated with significantly altered DMR and DGE are indicated in the figure. (TIF)

## Acknowledgments

The sequencing service was provided by the Norwegian Sequencing Centre (www.sequencing.uio.no), a national technology platform hosted by Oslo University Hospital and the University of Oslo supported by the Research Council of Norway and the Southeastern Regional Health Authority. We would like to thank Professor Christian Steinhäuser and Ph.D. Peter Bedner from the Institute of Cellular Neurosciences University of Bonn Medical Center for their help in establishing and traineeship on the animal model, consistent advice and friendship. We would like thank Professor Frank Kirchhoff for his excellent leadership of the EU Glia PhD consortium. We would further like to thank Ph.D. Hans Christian D. Aass (The Flow Cytometry Core Facility, Department of Medical Biochemistry, Oslo University Hospital, Oslo, Norway) for sorting nuclei. We would also like to thank Ph.D. Rune Enger (Glia Lab and Letten Centre, Department of Molecular Medicine, Division of Physiology, Institute of Basic Medical Sciences, University of Oslo, Oslo, Norway) for providing graphical visualizations of hippocampi used in S1 Fig. Finally, we would like to thank Øystein Horgmo (Medical Photography and Illustration Service, University of Oslo, Norway) for providing support regarding the graphical presentation. Parts of S1 Fig were modified from images provided by https://smart.servier.com/ under a Creative Commons Attribution 3.0 Unported License.

## Author Contributions

**Conceptualization:** Toni C. Berger, Magnus D. Vigeland, Hanne S. Hjorthaug, Lars Etholm, Erik Taubøll, Kjell Heuser, Kaja K. Selmer.

**Data curation:** Toni C. Berger, Magnus D. Vigeland.

**Formal analysis:** Toni C. Berger, Magnus D. Vigeland.

**Funding acquisition:** Kjell Heuser, Kaja K. Selmer.

**Investigation:** Toni C. Berger, Hanne S. Hjorthaug, Cecilie G. Nome, Kjell Heuser.

**Methodology:** Toni C. Berger, Magnus D. Vigeland, Hanne S. Hjorthaug, Lars Etholm, Erik Taubøll, Kjell Heuser, Kaja K. Selmer.

**Project administration:** Toni C. Berger, Magnus D. Vigeland, Hanne S. Hjorthaug, Erik Taubøll, Kjell Heuser, Kaja K. Selmer.

**Resources:** Erik Taubøll, Kjell Heuser, Kaja K. Selmer.

**Software:** Magnus D. Vigeland.

**Supervision:** Magnus D. Vigeland, Erik Taubøll, Kjell Heuser, Kaja K. Selmer.

**Validation:** Toni C. Berger, Magnus D. Vigeland, Kjell Heuser.

**Visualization:** Toni C. Berger, Magnus D. Vigeland.

**Writing – original draft:** Toni C. Berger, Magnus D. Vigeland, Hanne S. Hjorthaug, Kjell Heuser.

**Writing – review & editing:** Toni C. Berger, Magnus D. Vigeland, Hanne S. Hjorthaug, Lars Etholm, Cecilie G. Nome, Erik Taubøll, Kjell Heuser, Kaja K. Selmer.

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
