## [Decision Letter · Decision Letter 0]

10 Sep 2019

PONE-D-19-20265

Neuronal and glial DNA methylation and gene expression changes in early epileptogenesis

PLOS ONE

Dear Dr. Berger,

Thank you for submitting your manuscript to PLOS ONE. After careful consideration, we feel that it has merit but does not fully meet PLOS ONE’s publication criteria as it currently stands. Therefore, we invite you to submit a revised version of the manuscript that addresses the points raised during the review process.

We would appreciate receiving your revised manuscript by Oct 25 2019 11:59PM. To enhance the reproducibility of your results, we recommend that if applicable you deposit your laboratory protocols in protocols.io, where a protocol can be assigned its own identifier (DOI) such that it can be cited independently in the future. For instructions see: http://journals.plos.org/plosone/s/submission-guidelines#loc-laboratory-protocols

We look forward to receiving your revised manuscript.

Kind regards,

Giuseppe Biagini, MD

Academic Editor

PLOS ONE

Journal Requirements:

Reviewers' comments:

Reviewer's Responses to Questions

**Comments to the Author**

1. Is the manuscript technically sound, and do the data support the conclusions?

Reviewer #1: Yes

Reviewer #2: Partly

2. Has the statistical analysis been performed appropriately and rigorously? 

Reviewer #1: Yes

Reviewer #2: Yes

3. Have the authors made all data underlying the findings in their manuscript fully available?

Reviewer #1: Yes

Reviewer #2: No

4. Is the manuscript presented in an intelligible fashion and written in standard English?

Reviewer #1: Yes

Reviewer #2: Yes

5. Review Comments to the Author

Reviewer #1: Manusript by Berger et al. describes alterations in DNA methylation and gene expression in neurons and glia separately at 24h fllowing status epilepticus. Subject is very timely and of interest for many readers. Data are presented in details which enables readers to dig into particular molecular events of interest for them. I do not have any serious critical comments.

Minor issues:

- do animals in intracortical KA model develop epilepsy?

- tables 1-2 may be misleading since cells in rows do not present the same or similar categories but from different databases as it might be interpreted by reader

- does the method used by authors allow maintaining cell integrity during homogenization of frozen tissue, or is cytoplasm lost and only morre resistant nuclei are sorted?

Reviewer #2: Review to manuscript PONE-D-19-20265

In the present study, Berger et al. describe cell population specific DNA methylation and gene expression in mice (n=8 per group, KA and sham respectively), 24h following status epilepticus from intracortical (juxta-hippocampal) Kainic acid injection. FANS was used to sort NeuN-positive and NeuN-negative nuclei from pooled HC. DNA and total RNA were extracted and used for sequencing. The experimental design and data are mainly well described, however the following comments remain:

1. Neuronal and non-neuronal nuclei were isolated, not whole cells. NeuN-negative nuclei, do not necessarily have to originate from glial cells. As nuclei were sorted and analyzed not all transcription was studied, but only immediate transcripts that were still present in the nucleus. The preparation method was also not mRNA-specific, but total RNA was extracted. The reviewer is aware of the technical limitations for single cell-sequencing from adult brain tissue and thus only requests that the authors are specific about their methodology to not confuse the reader.

2. p9: „For any gene, its promoter region was defined as the 1 kb segment immediately upstream of the transcription start site and the upstream region from -5 kb to -1 kb, where negative numbers indicate positions upstream of transcription start site.“ Unclear. So the promoter definition was -1kb from TSS and the upstream region is -1kb to -5kb? Please explain why the „upstream region“ was considered interesting? What function it is supposed to have in gene regulation?

3. Authors should make sequencing data (raw files) available through GEO.

4. The authors state “We detected single genetic loci in several epilepsy-related genes, where DNAm and GE changes coincide. These may serve as potential target sites for epigenetic antiepileptogenic therapeutic intervention.” However, the number of differentially methylated CpGs/regions that correlated with gene expression was very low. Also the time point 24h after SE is very short. At this stage the acute response to the KA treatment and status is visible, which has little to do with the later epileptogenic process and questions the suitability of suggested targets for therapeutic interventions in chronic epilepsy. In general, this reviewer feels that the discussion of single genes from the present data overestimates the results. Rather, a discussion of technical limitations should be included.

5. Much information appears to be hidden in the supplements.

6. PLOS authors have the option to publish the peer review history of their article (what does this mean?). If published, this will include your full peer review and any attached files.

Reviewer #1: No

Reviewer #2: No

---

## [Author Response · Author response to Decision Letter 0]

30 Sep 2019

Manuscript number: PONE-D-19-20265

Title: Neuronal and glial DNA methylation and gene expression changes in early epileptogenesis

Dear Editor Professor Biagini, 

We thank you for the expert reviews on our paper!

Based on the helpful comments of the reviewers, we have now prepared a revised and expanded version of our manuscript. In our opinion, the revision has improved our paper and we trust that you will find the new version suitable for publication in PLOS ONE. 

Below are our responses to the reviewers’ comments.

Reviewer #1: Manuscript by Berger et al. describes alterations in DNA methylation and gene expression in neurons and glia separately at 24h following status epilepticus. Subject is very timely and of interest for many readers. Data are presented in detail which enables readers to dig into particular molecular events of interest for them. I do not have any serious critical comments.

Minor issues:

- do animals in intracortical KA model develop epilepsy?

We want to thank reviewer 1 for the positive review. 

The answer to the question from reviewer 1 is YES. More than 90% of animals treated with intracortical kainic acid develop epilepsy, as characterized in detail by Bedner P. et al., in the journal Brain in 2015. This is further confirmed by our own experience with the model (Szokol K. et al., Front Exp Neurosci, 2015). Accordingly, a more thorough description of the KA model has been added in the revised manuscript (methods). 

- tables 1-2 may be misleading since cells in rows do not present the same or similar categories but from different databases as it might be interpreted by reader

We agree with reviewer 1 and changed the tables 1 and 2 (now tables 3 and 4) accordingly. We think that they now provide a better overview of GO- and KEGG terms in neuronal and glial up- and downregulated genes. 

- does the method used by authors allow maintaining cell integrity during homogenization of frozen tissue, or is cytoplasm lost and only more resistant nuclei are sorted?

Our method represents a modified variant of Jiang et al., BMC Neurosci, 2008 and is based on sorting of cell nuclei, which has several advantages over cell based sorting (e.g. as stated nuclei are more resistant than whole cells; Grindberg R. et al., PNAS, 2013). Consequently, cytoplasm and its contents are lost. This is already specified in the methods section and has now also been revised in the abstract as well as added to the new technical limitations chapter. 

Reviewer #2: Review to manuscript PONE-D-19-20265

In the present study, Berger et al. describe cell population specific DNA methylation and gene expression in mice (n=8 per group, KA and sham respectively), 24h following status epilepticus from intracortical (juxta-hippocampal) Kainic acid injection. FANS was used to sort NeuN-positive and NeuN-negative nuclei from pooled HC. DNA and total RNA were extracted and used for sequencing. The experimental design and data are mainly well described, however the following comments remain:

1. Neuronal and non-neuronal nuclei were isolated, not whole cells. NeuN-negative nuclei, do not necessarily have to originate from glial cells.

As nuclei were sorted and analyzed not all transcription was studied, but only immediate transcripts that were still present in the nucleus. 

The preparation method was also not mRNA-specific, but total RNA was extracted. The reviewer is aware of the technical limitations for single cell-sequencing from adult brain tissue and thus only requests that the authors are specific about their methodology to not confuse the reader.

First, we want to thank reviewer 2 for the in-depth review. 

Reviewer 2 is right regarding the NeuN- fraction containing a small number of other cell types than glia. In the revised manuscript, we now have specified this in a new section on technical limitations. 

Regarding RNA sequencing, we first isolated total RNA from sorted nuclei. The consecutive step of using a oligo(dT) primer for cDNA creation sub selects mRNA transcripts. We agree with reviewer 2 that this has not been pointed out clearly in the article. We have now adjusted nomenclature and added details in the methods section of the revised manuscript for clarification (methods, supplementary methods). Concerning the use of nuclear mRNA, we are aware of technical limitations like enrichment of mRNA coding for proteins with nuclear functions (Barthelson RA et al., BMC Genomics, 2007) and a potentially lower level of immediate early genes (Bakken TE et al., PLOS ONE, 2018) compared to cytosolic mRNA. This has now been added to the discussion (technical limitations). 

2. p9: „For any gene, its promoter region was defined as the 1 kb segment immediately upstream of the transcription start site and the upstream region from -5 kb to -1 kb, where negative numbers indicate positions upstream of transcription start site.“ Unclear. So the promoter definition was -1kb from TSS and the upstream region is -1kb to -5kb? Please explain why the „upstream region“ was considered interesting? What function it is supposed to have in gene regulation?

We thank the reviewer for the insightful feedback. Regarding the definition of genomic regions, we have modified our formulations in the paper (methods). As correctly stated, the upstream region was defined as from -5kb to -1kb and promoter region from -1kb to TSS.

Concerning the use of the upstream region as a separate genomic feature, we used predefined genomic feature types supplied by the annotatr package in R, having no à priori hypothesis of possible importance of genomic feature types per se. 

3. Authors should make sequencing data (raw files) available through GEO.

RAW files have now been uploaded to GEO (GSE138100). 

4. The authors state “We detected single genetic loci in several epilepsy-related genes, where DNAm and GE changes coincide. These may serve as potential target sites for epigenetic antiepileptogenic therapeutic intervention.” However, the number of differentially methylated CpGs/regions that correlated with gene expression was very low. Also the time point 24h after SE is very short. At this stage the acute response to the KA treatment and status is visible, which has little to do with the later epileptogenic process and questions the suitability of suggested targets for therapeutic interventions in chronic epilepsy. 

Several animal models of temporal lobe epilepsy have shown that important cellular/molecular changes already occur during the first hours and days after status epilepticus (Bedner P. et al., Brain, 2015; Rakhade SN et Jensen FE, Nat Rev Neurol. 2009). In this model, the latent phase of epilepsy starts after around 4 hours (post injection) and lasts 5+-2.9 days (see also figure below). We therefore believe that 24 hours post injection is an appropriate time point for studying early epigenetic mechanisms and gene expression precipitating downstream molecular changes. We agree with reviewer 2 that it also would be of high interest to study DNA methylation and gene expression at a later time point reflecting a chronic stage of epileptogenesis. 

Figure: Epileptogenesis in the intracortical kainic acid model of mTLE (modified from Bedner P. et al., Brain, 2015)

In general, this reviewer feels that the discussion of single genes from the present data overestimates the results. Rather, a discussion of technical limitations should be included.

We are aware of the limitations and restricted interpretability of a few genomic loci at one time point and have revised the discussion accordingly as well as added a new section on technical limitations. We want to thank reviewer 2 for the critical feedback, as this surely will increase the quality of this paper.

5. Much information appears to be hidden in the supplements.

We have integrated the list of differentially expressed genes (up- and downregulated genes) as Tables 1 and 2 into the revised manuscript (numbers of the previous tables have been adjusted).

With kind regards, 

Toni Berger

---

## [Decision Letter · Decision Letter 1]

20 Nov 2019

PONE-D-19-20265R1

Neuronal and glial DNA methylation and gene expression changes in early epileptogenesis

PLOS ONE

Dear Dr. Berger,

Thank you for submitting your manuscript to PLOS ONE. After careful consideration, we feel that it has merit but does not fully meet PLOS ONE’s publication criteria as it currently stands. Therefore, we invite you to submit a revised version of the manuscript that addresses the following points:

- Indicate how animals were killed.

- Provide a section describing statistics in methods.

- Check tables for values' unit (thousands).

We would appreciate receiving your revised manuscript by Jan 04 2020 11:59PM. To enhance the reproducibility of your results, we recommend that if applicable you deposit your laboratory protocols in protocols.io, where a protocol can be assigned its own identifier (DOI) such that it can be cited independently in the future. For instructions see: http://journals.plos.org/plosone/s/submission-guidelines#loc-laboratory-protocols

We look forward to receiving your revised manuscript.

Kind regards,

Giuseppe Biagini, MD

Academic Editor

PLOS ONE

Reviewers' comments:

Reviewer's Responses to Questions

**Comments to the Author**

1. If the authors have adequately addressed your comments raised in a previous round of review and you feel that this manuscript is now acceptable for publication, you may indicate that here to bypass the “Comments to the Author” section, enter your conflict of interest statement in the “Confidential to Editor” section, and submit your "Accept" recommendation.

Reviewer #1: All comments have been addressed

Reviewer #2: All comments have been addressed

2. Is the manuscript technically sound, and do the data support the conclusions?

Reviewer #1: Yes

Reviewer #2: Yes

3. Has the statistical analysis been performed appropriately and rigorously? 

Reviewer #1: Yes

Reviewer #2: Yes

4. Have the authors made all data underlying the findings in their manuscript fully available?

Reviewer #1: Yes

Reviewer #2: Yes

5. Is the manuscript presented in an intelligible fashion and written in standard English?

Reviewer #1: Yes

Reviewer #2: Yes

6. Review Comments to the Author

Reviewer #1: no further comments

Reviewer #2: (No Response)

7. PLOS authors have the option to publish the peer review history of their article (what does this mean?). If published, this will include your full peer review and any attached files.

Reviewer #1: No

Reviewer #2: No

---

## [Author Response · Author response to Decision Letter 1]

25 Nov 2019

Dear Editor Professor Biagini, 

We thank you for your feedback regarding our paper: 

 - Indicate how animals were killed.

 - Provide a section describing statistics in methods.

 - Check tables for values' unit (thousands).

All three points have been addressed and the manuscript adjusted accordingly. 

We hope that you will find the new version suitable for publication in PLOS ONE. 

With kind regards, 

Toni Berger

---

## [Editor Report · Decision Letter 2]

3 Dec 2019

Neuronal and glial DNA methylation and gene expression changes in early epileptogenesis

PONE-D-19-20265R2

Dear Dr. Berger,

We are pleased to inform you that your manuscript has been judged scientifically suitable for publication and will be formally accepted for publication once it complies with all outstanding technical requirements.

With kind regards,

Giuseppe Biagini, MD

Academic Editor

PLOS ONE
---

## [Editor Report · Acceptance letter]

10 Dec 2019

PONE-D-19-20265R2 

Neuronal and glial DNA methylation and gene expression changes in early epileptogenesis 

Dear Dr. Berger:

I am pleased to inform you that your manuscript has been deemed suitable for publication in PLOS ONE. Congratulations! Your manuscript is now with our production department. 

With kind regards,

on behalf of

Dr. Giuseppe Biagini 

Academic Editor

PLOS ONE